# Unsupervised discovery of symmetries and symmetry-based domains from raw data

## Abstract

Signals are often generated by processes that respect a symmetry group of the domain they live on. Observed signals are obtained from underlying samples by some transformation corresponding to a physical process, resulting in a group action that is often more complicated than the action on the underlying domain. Learning the symmetry group and the underlying symmetry-based domain are two intertwined problems of fundamental importance. In this paper, we develop a method that simultaneously discovers symmetries and symmetry-based domains in a fully unsupervised setting, without assuming that the group action is transitive. Our approach is based on a lifting operation inspired by Group Convolutional Networks, mapping the space of observed features to a domain parametrized by group elements. By utilizing a powerful locality prior, we are able to learn symmetry actions such as translations, permutations and frequency shifts, on datasets with much higher dimensionalities than has been possible before. Since the domain is hidden, we assume the symmetry group acts directly on the space of samples, which in the familiar case of natural images means the underlying pixel translation symmetries to be learned are for a set of images. As well as discovering the relevant symmetries directly from raw data, our method also offers a new approach towards solving linear inverse problems. Our code can be found at Github.

## 1 Introduction

The importance of symmetries of data distributions and symmetry-based representations is widely recognized (Bronstein et al., 2021; Anselmi et al., 2019; Anselmi & Patel, 2022; Higgins et al., 2022; Tonnaer et al., 2023; 2020; Godfrey et al., 2023; Perin & Deny, 2025; Olanrewaju, 2025). In some cases such as time series data with time translation symmetry, the action of the relevant symmetry group and the symmetry-based representation on the natural *domain* of data are known a priori, but the discovery of symmetries and the domains where the symmetries manifest remains mostly an unsolved problem.

Current methods for symmetry discovery are either scalable but require structured input such as sequential representations related by actions of symmetry group elements (Greydanus et al., 2019; Alet et al., 2021; Churchill & Xiu, 2023; Sohl-Dickstein et al., 2010; Dehmamy et al., 2021; Koyama et al., 2023; Miyato et al., 2022; Park et al., 2022; Mitchel et al., 2024; Hou et al., 2024), or are limited to very low dimensional group representations (Desai et al., 2022; Tombs & Lester, 2022; Yang et al., 2023b;a). The unsupervised discovery of symmetries and symmetry-based representations from high-dimensional data without assuming a known domain (in the sense of (Bronstein et al., 2021)) *and* without having access to samples related to each other by symmetry operations, remains a challenge.

The prototypical example of such a problem is the blind discovery of the underlying translation symmetry of image data. Given a set of natural images turned into vectors, can one discover the generators of pixel translations, without access to samples related by translations? Even for images of $10 \times 10$ pixels, this would entail a search for $100 \times 100$ matrices representing symmetry generators, and this search is fraught with difficulties, not the least being a precise formulation of the problem in the first place.

Figure 1: Training snapshots from the permuted-translation symmetry experiment. The left panel shows the model's input: a bag-of-pixels representation created by randomly shuffling the pixels of an MNIST digit, thereby destroying its spatial structure. The right panel demonstrates how the model learns the inverse transformation, progressively restoring the digit and recovering a translation-invariant representation. To ease visual interpretation, we transposed the entire image.

In this study, we combine the problems of symmetry discovery and symmetry-based domain discovery (and signal reconstruction) in the relatively modest setting of abelian Lie groups. By performing a simultaneous search for both the symmetries and the symmetry-based domain using an information-theoretic approach, we are able to discover the symmetry group and reconstruct signals in the hidden domain for a challenging set of examples. We demonstrate the efficacy of the method by discovering the matrices representing pixel translations from a set of raw images (no data augmentation, no explicitly translated pairs of samples), finding the neighborhood relation and the symmetry generators in a shuffled case of the Ising model from statistical physics (Külske, 2025), finding the frequency-shift operator for a synthetic time series dataset that is invariant under frequency shifts, and other related examples. For cases where the domain is hidden (such as shuffled pixels of images), we demonstrate that the method is able to discover (unshuffle) the hidden signals while simultaneously finding the relevant symmetry matrices (see Figure 1 ). All these problems were previously inaccessible in the fully unsupervised setting.

An essential component of our method is the **lifting convolution**, adopted from Group Convolutional Networks (GCNs) (Cohen & Welling, 2016; Bekkers, 2021; Romero & Lohit, 2021; Finzi et al., 2020; Kondor & Trivedi, 2018) and tailored to unsupervised symmetry discovery. Upon training by minimizing an information-theoretic loss function, this operation maps the input space where symmetries are possibly obscure to a new representation space where the symmetries act as simple translations. The method does *not* assume that the symmetry group acts on the set of possible examples transitively. While transitive group actions are fundamental, they do not cover many real-world examples. A dataset consisting of a given image and all its rotated versions has the rotation group acting on it transitively, but for a general set of images with an underlying rotation symmetry (acting on the distribution), it is not possible to obtain an arbitrary image by rotating a given reference image.

In many physical settings, one does not observe the signal generating mechanism directly, but instead sees the system through the lens of an observational procedure, or a signal-modifying medium. While the underlying physical system may have manifest symmetries coupled to the locality of the signal generation, in the observed versions, the symmetries may be obscure, and the signals may be distorted. Thus, an unsupervised method for the discovery of symmetries *and* the representation of the signals in the hidden domains is, in a sense, closely related to a wide range of inverse problems encountered in signal processing. We believe our work makes a modest but solid contribution to the problem of solving linear inverse problems and extracting unobfuscated, natural signal representations by learning their hidden symmetric structure.

We summarize our core contributions as follows:

**Ability to discover the symmetries when they act intransitively:** Via using a lifting convolution inspired from GCNs, our method can discover symmetries when they act each constituent forming the dataset in different ways. This overcomes a key limitation in unsupervised symmetry-discovery literature. To formalize briefly, our method can discover symmetries when the group acts **intransitively**.

**Discovery of symmetry-based domains:** We demonstrate that our framework can successfully *infer the underlying, original signals from distorted input data*. This capability opens new horizons for tackling challenging unsupervised inverse problems.

**Scalability to high-dimensional representations:** By leveraging *locality prior*, our method scales to discover symmetry group representations (irreps) with dimensionalities as high as $\mathbf{27^2 \times 27^2}$, overcoming the scalability barriers of probabilistic inference at high dimensional spaces.

**Generalized symmetry representations:** Our framework learns representations that act on the space of all possible samples, *moving beyond the limitations of left-regular representations* that are confined to predefined coordinates.

## 2 RELATED WORK

Various studies explored supervised-learning-based approaches to symmetry discovery. Some examples include symmetry searches via discovering augmentations of image datasets (Benton et al., 2020), (Romero & Lohit, 2021), seeking transformations that leave predetermined or learned functions invariant (Forestano et al., 2023a;b; Krippendorf & Syvaeri, 2020; Moskalev et al., 2022; Ko et al., 2024) and methods for learning weight-sharing mechanisms that give the best performance for a group of supervised-learning tasks (Zhou et al., 2020). Although this line of research often results in robust methods, their task-specificity can also be a limitation; symmetry candidates that are suited to one supervised learning task is often not appropriate for another, and the requirement of a labeled dataset is not always satisfied.

A closely related avenue of research is the discovery of the time evolution of dynamical systems (Greydanus et al., 2019; Alet et al., 2021; Churchill & Xiu, 2023) and evolutions of sequential data (Sohl-Dickstein et al., 2010; Dehmamy et al., 2021; Koyama et al., 2023; Miyato et al., 2022; Park et al., 2022; Mitchel et al., 2024). While this auto-regressive or self-supervised setting provide powerful methods without requiring labeled data in the usual sense, they require domains satisfying translational-symmetry or sequential data to operate. We are rather focused over discovering symmetries coupled to domains. In this manner, methods proposed in this line of research has complementary nature to our method.

In similar spirit to our method, the methods based on distribution invariance also don't require structured representation to operate as well as don't requiring labeled datasets (Desai et al., 2022; Tombs & Lester, 2022; Yang et al., 2023b;a). They are demonstrated to work with dimensional irreducible representations up-to $4 \times 4$ dimensions acting over either position coordinates or phase-space coordinates. For applying their method at higher dimensions, it's assumed that the symmetry-representation is in block diagonal form, and the blocks are identical with each other Yang et al. (2023a).

Although methods based on probabilistic invariance is a source of inspiration for us, scalability problems hinder their applicability to raw data since probabilistic invariance suffers from curse of dimensionality, and they have been reported to have convergence failures at $7 \times 7$ dimensional representations (Efe & Ozakin, 2024). The solution to address this problem, assuming that symmetry representation acts in a block-diagonal form with identical blocks Yang et al. (2023a), is a major limitation in our setting.

In this study, we address a fundamental complementary problem which is not yet addressed to our knowledge. We are interested in the symmetry representations that form the domain, such as translation representations, and this requires addressing different challenges.

## 3 BACKGROUND

### 3.1 SYMMETRIES OF THE DATA DISTRIBUTION

We represent the data by a random vector $\mathbf{X}$ with distribution $p_X$ on a space $\mathbb{X}$. We say that the distribution of $\mathbf{X}$ is invariant under the action of a group $G$ with representation $\rho : G \to \mathrm{GL}(\mathbb{X})$ if

$$p_X(\rho(g)\mathbf{x}) = p_X(\mathbf{x}) \qquad \forall g \in G, \mathbf{x} \in \mathbb{X}.$$

Our aim is to discover the symmetry group $G$, and to use it for building a symmetry-based representation revealing the true nature of data.

## 3.2 DATA MODEL AND SIGNAL RECOVERY

**Data model:** We model the underlying process generating the data as a stochastic process on a Lie group $G$, which also forms the hidden, symmetry-based domain of the data—in the notation of (Bronstein et al., 2021) $\Omega = G$. Each sample on this hidden domain is thus a function $f : G \to \mathbb{R}$, which we think of as an element of a vector space of functions defined on $G$, e.g., functions that are square integrable with respect to the Haar measure on $G$. To state formally, we consider that the signals $f$ live in the space $\mathcal{F}$ given by $\mathcal{F}(\Omega, \mathcal{C}) = L^2(G, \mathbf{R})$.

We assume the **process generating these samples is invariant (stationary) under the action of** $G$ **via left translations**, in other words, under $(v \cdot f)(u) := f(v^{-1} \cdot u)$ where $u, v \in G$. We will take $G$ to be abelian, and write the group operation as addition, which gives $(v \cdot f)(u) := f(u - v)$.

We consider that the *observed* signals are not the functions $f$, but are vectors in the vector space $\mathbb{X}$, (which we will take to be a Hilbert space), which are obtained from the underlying samples $f$ on $G$ via a linear transformation $\mathcal{M} : \mathcal{F} = L^2(G, \mathbb{R}) \to \mathbb{X}$, $\mathcal{M} : f \mapsto \mathbf{x} \in \mathbb{X}$. This transformation is to be thought of as relating underlying physical samples to the observed samples via a physical process such as measurement itself.

We assume that the action of $G$ on underlying signals $f$ defines a unitary group representation $\rho$ on $\mathbb{X}$, i.e.,

$$\mathcal{M}[v \cdot f] = \rho(v) \cdot \mathcal{M}[f], \tag{1}$$

where $\rho \in GL(\mathbb{X})$ is a linear map $\mathbb{X} \to \mathbb{X}$.

Our aim is to learn the symmetry group action $\{\rho(g) : g \in G\}$ on the space of observed samples $\mathbb{X}$, and to learn the underlying domain $G$, together with a map that reconstructs the underlying signals $f$ on $G$ from observed signals $\mathbf{x}$. Note that this assumes $\mathcal{M}$ is an invertible map between the space $\mathcal{F}$ of possible underlying samples and the space of possible observations.

**Recovering hidden symmetry-based signals:** To describe the process of recovering the hidden, symmetry-based signals from observed ones, we start with the notion of a Dirac delta function $\delta$ on $G$ satisfying,

$$f(u) = \int_G f(v)\delta(v^{-1}u)d\mu(v), \tag{2}$$

where $d\mu$ is the Haar measure on $G$. For the case of an abelian Lie group $G$ of dimension $P$ with Lie algebra $\mathfrak{g}$, we denote a basis for the Lie algebra $\mathfrak{g}$ of $G$ by $\mathfrak{l}_j$, $j = 1, \ldots, P$. The exponential map allows us to label the group elements by a set of real numbers $\mathbf{t} = t_1, \ldots t_P$, $g(\mathbf{t}) = \exp(\sum_j \mathfrak{l}_j t_j)$. The corresponding matrix $\rho(\mathbf{t})$ becomes $\rho(\mathbf{t}) = \exp(\sum_j L_j t_j)$ where $L_j$ is the matrix representing the Lie algebra element $\mathfrak{l}_i$. When $\rho$ is unitary, the $L_j$ are anti-Hermitian. In the special case of an orthogonal representation, $L_j$ is anti-symmetric. In the case of abelian $G$, we parametrize the group using the parameters $t_j$ and with an appropriate choice of normalization for $d\mu$, we write the integral (2) as

$$f(\mathbf{t}) = \int f(\mathbf{s})\delta(\mathbf{t} - \mathbf{s})d\mathbf{s}. \tag{3}$$

We will call the image of $\delta$ under the action of $\mathcal{M}$ the *origin vector*, and denote it by $\boldsymbol{\delta}_o = \mathcal{M}[\delta]$. Using (1), we get $\mathcal{M}[v \cdot \delta] = \rho(v) \cdot \boldsymbol{\delta}_o$.

Just as any signal on $G$ can be formed by a linear combination of delta functions via (2), any observed signal can be written as a linear combination of appropriately transformed versions of $\boldsymbol{\delta}_o$:

$$\mathbf{x} = \mathcal{M}[f] = \mathcal{M}\left[\int_G f(v)(v \cdot \delta)d\mu(v)\right] = \int_G f(v)\mathcal{M}[v \cdot \delta]d\mu(v) = \int_G f(v)\rho(v) \cdot \boldsymbol{\delta}_o d\mu(v). \tag{4}$$

Note that the last integral is vector-valued.

Our aim is to learn $\rho(g)$ and $\boldsymbol{\delta}_o$ from data, and using this knowledge, reconstruct $f$, the underlying, symmetry-based representation of the signal. We next show that $f$ can indeed be reconstructed using

a standard deconvolution approach under certain assumptions. Consider the function $z : G \to \mathbb{R}$ defined by projecting an observed signal onto the orbit generated by the group $G$ acting over origin $\boldsymbol{\delta_o}$:

$$z(\mathbf{t}) := {\boldsymbol{\delta_o}}^\top \rho(-\mathbf{t})\mathbf{x} \quad = {\boldsymbol{\delta_o}}^\top \rho(-\mathbf{t}) \left[ \int f(\mathbf{s})\rho(\mathbf{s}) \cdot \boldsymbol{\delta_o} d\mathbf{s} \right] = \int f(\mathbf{s})[{\boldsymbol{\delta_o}}^\top \rho(\mathbf{s} - \mathbf{t}) \cdot \boldsymbol{\delta_o}]d\mathbf{s}$$

$$= (f * k)(-\mathbf{t})$$

where we defined the kernel $k(\mathbf{r}) := {\boldsymbol{\delta_o}}^\top \rho(\mathbf{r}) \cdot \boldsymbol{\delta_o}$ and used the underlying, symmetry-based representation of the signals $f : \mathbb{R}^P \to \mathbb{R}$. Solving $f$ from $z$ is a deconvolution problem, whose solution is outlined for the case of a compact $G$ in the Appendix A.1.

As a result of the deconvolution process we obtain $f(t) = \boldsymbol{\phi}^\top \rho(-t)\mathbf{x}$. This equation tells us that we can recover the hidden signals on the hidden domain by applying an appropriate projection defined by the resolution filter, coupled with the action of the group elements. In the Group Convolutional Network literature, a closely related operation is already defined and named as **lifting convolution** (Bekkers, 2021; Romero & Lohit, 2021). Below, we will also consider the case of the lifting operation for a general $\boldsymbol{\phi}$, possibly different from (20) combined with the correct $\rho(t)$.

Formally, for Abelian Lie Groups, we define the lifting operation $\mathcal{L} : \mathbb{X} \to L^2(\mathbb{R}^P)$ as;

$$(\mathcal{L}\mathbf{x})(\mathbf{t}) := \boldsymbol{\phi}^\top \rho(-\mathbf{t})\mathbf{x} = f(\mathbf{t}) \tag{5}$$

mapping samples from the space of observed signals to hidden, symmetry-based representations.

### 3.3 PROPERTIES OF THE LIFTING OPERATION

We next discuss some properties of the lifting operation for a general $\boldsymbol{\phi}^\top$ (not necessarily the one defined by (20)).

**Mapping samples to continuous functions:** Let $\mathbf{x}$ be an observed sample, and $\mathbf{t}_1, \mathbf{t}_2 \in \mathbb{R}^P$ two points in the group parameter space. Then, the values $y_1 := \mathcal{L}\mathbf{x}(\mathbf{t}_1)$ and $y_2 := \mathcal{L}\mathbf{x}(\mathbf{t}_2)$ of the lifted function at $\mathbf{t}_1$ and $\mathbf{t}_2$ cannot be too different when $\mathbf{t}_1$ and $\mathbf{t}_2$ are close. Namely, one has

$$|y_1 - y_2| = |\boldsymbol{\phi}^\top \rho(-\mathbf{t}_1)\mathbf{x} - \boldsymbol{\phi}^\top \rho(-\mathbf{t}_2)\mathbf{x}| \leq ||\boldsymbol{\phi}^\top|| \, ||\rho(-\mathbf{t}_1) - \rho(-\mathbf{t}_2)|| \, ||\mathbf{x}|| \tag{6}$$

where $||\rho(-\mathbf{t}_1) - \rho(-\mathbf{t}_2)||$ denotes a suitable norm. In the practical case of finite-dimensional vectors $\mathbf{x}$, $\rho$ will be a matrix, and one can take the Frobenius norm. The continuity of the exponential map ensures that $||\rho(\mathbf{t}_1) - \rho(\mathbf{t}_2)||_F \to 0$ when $\mathbf{t}_1 \to \mathbf{t}_2$, showing that the lifting operation maps $\mathbf{x}$ to a continuous function over group elements.

**Equivariance:** However, even when $\boldsymbol{\phi}^\top$ is not the correct resolving filter defined by (20), the lifting map is equivariant under the action of $G$. We have,

$$(\mathcal{L}\rho(\mathbf{s}) \cdot \mathbf{x})(\mathbf{t}) = \boldsymbol{\phi}^\top \rho(-\mathbf{t})\rho(\mathbf{s})\mathbf{x} = \boldsymbol{\phi}^\top \rho(-(\mathbf{t} - \mathbf{s}))\mathbf{x} = (\mathcal{L}\mathbf{x})(\mathbf{t} - \mathbf{s}) . \tag{7}$$

Thus, one has $\mathcal{L} \circ \rho(\mathbf{s}) = T_\mathbf{s} \circ \mathcal{L}$ where $T_\mathbf{s}$ denotes the translation operator, acting on functions on $G$ by $(T_\mathbf{s} f)(\mathbf{t}) = f(\mathbf{t} - \mathbf{s})$.

**Converting group invariance to shift-invariance:** One of the essential properties of the proposed lifting operation is that it converts the symmetries of the data distribution into a much more tractable shift-invariance. If the data distribution $p_X$ (for observed signals) is $\rho$-invariant, then the lifting $\mathcal{L}\mathbf{x}(t) = \boldsymbol{\phi}^\top \rho(-\mathbf{t})\mathbf{x}$ is a stationary (shift-invariant) random field on $G$ (for proof, see Appendix A.2).

In the case where $\boldsymbol{\phi}^\top$ is given by (20) and the lifting operation is able to recover the correct underlying signals, $\hat{f} = f$, this relation simply says that the invariance of the distribution of observed signals under $\rho(\mathbf{t})$ is the same thing as the invariance of the underlying signals on $G$ under the left action of $G$.

## 4 METHODOLOGY

### 4.1 OVERVIEW

Our method involves two separate networks which work as a whole: the **lifting network** and the **auxiliary network**. The two networks are trained in two separate optimization loops, simultaneously.

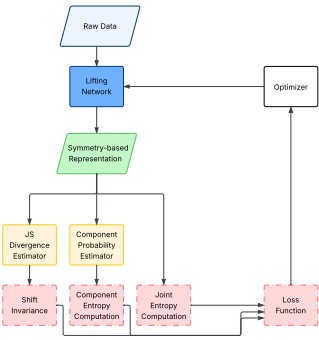
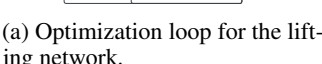
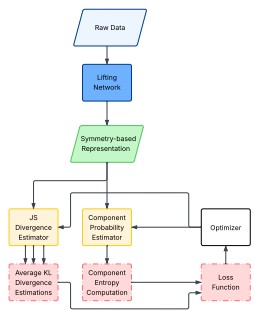
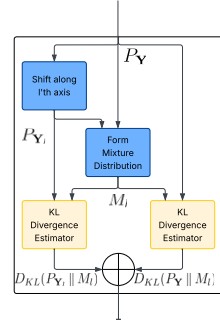

(a) Optimization loop for the lifting network.

(b) Optimization loop for the auxiliary networks.

(c) Structure of JS divergence estimator.

The lifting network consists of a single layer which implements the operation described in the Equation 5, in other words, it maps signals from the input space $\mathbb{X}$ to a candidate symmetry-based representation. It is trained using a loss function that consists of three pieces measuring the quality of the candidate symmetry-based representation, namely, the **shift-invariance**, **resolution** and **infomax** terms to be described below.

The auxiliary network contains subnetworks that estimate the loss terms of the lifting network. This involves estimating KL-divergences between the symmetry-based representation and its shifted versions, and estimating the overall entropy of the symmetry-based representation and the marginal entropies of its components. The auxiliary network has two loss terms whose optimization improves the quality of estimated KL-divergences and entropies.

## 4.2 LIFTING NETWORK

### 4.2.1 STRUCTURE

**Lie algebra basis parametrization** The lifting operation is performed using matrices $\{L_i\}_{i=1}^{P} \in \mathbb{R}^{d \times d}$ representing the action of the Lie algebra of the symmetry group on the space of observations. The lifting network parametrizes these matrices, which we call the Lie algebra basis, as real antisymmetric matrices which commute with each other, the antisymmetry ensuring the orthogonality of the representation $\rho$ via the exponential map. We ensure commutativity by expressing $L_i$ in terms of their eigendecomposition, using shared eigenvectors but allowing the eigenvalues to differ (see Appendix C.2). Thus, the network is tasked with learning the shared eigenvectors and the eigenvalues of $L_i$.

**Symmetry-based representation** Given $L_i$ and we form a discrete symmetry-based representation $y_{\mathbf{n}}$ by lifting the samples $\mathbf{x}$ via the operation

$$y_{\mathbf{n}} := (\mathcal{L}\mathbf{x})(\mathbf{t_n}) = \boldsymbol{\phi}^{\top}\rho(-\mathbf{t_n})\mathbf{x}. \tag{8}$$

Here, $\rho(-\mathbf{t_n})$ is the group element $\exp \sum_j L_i t_{\mathbf{n}_i}$, and $\mathbf{t_n}$ is the parameter vector specifying group elements, given as $\mathbf{t_n} = \left(\frac{n_1}{N_1}, \frac{n_2}{N_2}, \dots, \frac{n_P}{N_P}\right)$ where $N_i$ specifies the discretization resolution of $i$th group dimension. The resulting discrete **symmetry-based representation** $y_{\mathbf{n}}$ is thus of the form $y_{\mathbf{n}} : \mathbb{Z}^P \to \mathbb{R}$. We limit the integers $n_i$ to the interval $[-\frac{N_i}{2}, \frac{N_i}{2}]$. To form $y_{\mathbf{n}}$, the network needs to learn $L_i$ and $\boldsymbol{\phi}^{\top}$, which is simply a vector.

We implement this otherwise computationally expensive structure in an efficient manner by diagonalization based procedure. For details, see Appendix .

### 4.2.2 THE LIFTING LOSS

The loss for the lifting network is given by

$$\mathcal{L}_{\text{lifting}} = \mathcal{L}_{\text{invariance}} + \alpha \mathcal{L}_{\text{resolution}} + \beta \mathcal{L}_{\text{infomax}}.$$

Each term quantifies a different aspect of the quality of the symmetry-based representation obtained by lifting: the **invariance** term enforces shift invariance, the **resolution** term induces a notion of

locality, and the **information maximization** helps preserve information and avoid trivial solutions. All three terms are information-theoretic quantities with entropy-based formulations, which helps with hyperparameter tuning. We next describe these terms, assuming one has access to the relevant entropy-based quantities, which will be estimated by the auxiliary network.

**Invariance** The shift invariance loss is based on the Jensen-Shannon divergence $D_{JS}(P\|Q)$ between two distributions $P$ and $Q$, which is simply the symmetrized form of the KL divergence: $D_{JS}(P\|Q) = \frac{1}{2}\left(D_{KL}(P\|M) + D_{KL}(Q\|M)\right)$. For each group dimension $l = 1, \ldots, P$, we define the $l$-shifted version $\mathbf{y}^{(l)}$ of the symmetry-based representation by $y_{\mathbf{n}}^{(l)} := y_{\mathbf{n}+\mathbf{e}_l}$, where $\mathbf{e}_l \in \mathbb{R}^P$ denotes the $l$-th standard basis vector. In other words, we shift the $l$th component of a symmetry-based representation $\mathbf{y}$ by one unit to obtain $y^{(l)}$. Using capital letters to denote the corresponding random variables, we define the invariance loss as the average JS-divergence between $\mathbf{Y}$ and $\mathbf{Y}^{(l)}$:

$$\mathcal{L}_{\text{invariance}} := \frac{1}{P} \sum_{l=1}^{P} D_{\text{JS}}\left(P_{\mathbf{Y}}\|P_{\mathbf{Y}^{(l)}}\right), \tag{9}$$

where $P_{\mathbf{Y}}$ and $P_{\mathbf{Y}^{(l)}}$ denote the distributions of the random vectors $\mathbf{Y}$ and $\mathbf{Y}^{(l)}$, respectively. This quantity is minimized when the joint ($P$-dimensional) distribution of a candidate symmetry-based representation is invariant under shifts of its component indices $n_l$ by one. The estimation of $\mathcal{L}_{\text{invariance}}$ is described below.

**Resolution** To reveal the hidden real-world signals in an unsupervised manner, learning the symmetry group representation is not sufficient by itself, one also needs to learn the resolving filter. To do this, we use a loss term that forces independence across the components of the symmetry-based representation as much as possible. This is done via **total correlation** minimization Hyvärinen & Oja (2000) which is defined for $\mathbf{Y}$, the random vector representing the symmetry-based representation as

$$\mathcal{L}_{\text{resolution}} := \sum_{\mathbf{n}} h(Y_{\mathbf{n}}) - h(\mathbf{Y}) \tag{10}$$

where $h(Y_{\mathbf{n}})$ and $h(\mathbf{Y})$ are component-wise and joint entropies, respectively. This term is a powerful regularizer since it can probe any linear, non-linear, and higher-order dependency between the components of $\mathbf{Y}$. It will be shown in Section A.2.1 that due to the smoothness property of lifting (see Section 3.3), which forces close components being correlated, $\mathcal{L}_{\text{resolution}}$ induces information locality along the symmetry axes.

**Information maximization** To ensure that lifting preserves information and does not uncontrollably collapse the data, we maximize the joint entropy of the symmetry-based representation. In other words, we minimize

$$\mathcal{L}_{\text{infomax}} := -h(\mathbf{Y}) \tag{11}$$

In Section 4.3.2 below, we will describe an adaptive-rank approach to entropy estimation, which will lead this loss term to help the group orbits in $\mathbb{X}$ to initially span a small but high-entropy subspace, and then gradually expand it to span the whole space of possible signals. This procedure helps guide the optimizer along an effective trajectory, and prevents getting stuck in local minima. Minimizing $\mathcal{L}_{\text{infomax}}$ can also be seen as maximizing the mutual information between the input and the output of the lifting map (see the infomax principle (Linsker, 1988; Bell & Sejnowski, 1995)).

## 4.3 Auxiliary Network

### 4.3.1 KL Divergence Estimation

The invariance loss (9) is given in terms of the KL divergence between two distributions. To estimate KL divergence, we use the duality formula Sreekumar & Goldfeld (2022); Donsker & Varadhan (1975) giving the KL divergence $D_{KL}(R\|S)$ between two probability distributions $R, S$ as a solution to the optimization problem

$$D_{KL}(R\|S) = \sup_{\theta \in \Theta}\left\{\mathbb{E}_{\mathbf{x} \sim R}[f_\theta(\mathbf{x})] - \log \mathbb{E}_{\mathbf{x} \sim S}[\exp f_\theta(\mathbf{x})]\right\} \tag{12}$$

where $f_\theta : \mathbb{R}^N \to \mathbb{R}$ and $\theta$ parametrizes all functions Donsker & Varadhan (1975).

We approximate the search over all $f_\theta$ using a neural network, with $\theta$ describing the network parameters, and using (12) to define the loss. To approximate the expectation values in (12), we use averaging over batches. For the network architecture, we use a CNN with additional learnable position embeddings, which turns out to work rather efficiently. See Appendix C.5.2 for details. Since each JS-divergence involves two KL-divergence estimates, we use two identical networks with separate weights for each term in the loss (9), and train these networks using the optimization loop given in Figure 2b.

### 4.3.2 Entropy Estimation

To estimate the loss terms (11) and (10), we need to estimate the componentwise entropies and the joint entropy for the symmetry-based representation $\mathbf{Y}$.

**Component-wise Entropies** We estimate $\{h(Y_\mathbf{n})\}$ via Gaussian mixture-based trainable estimators as described in Pichler et al. (2022) where $Y_\mathbf{n}$ denotes the random variable representing single component of the symmetry-based representation. This approach estimates the parameters of a Gaussian mixture while using an entropy-based loss. We use 4 Gaussians per component of symmetry-based representation. See Appendix C.3 for details.

**Joint Entropy** We estimate the joint entropy of the symmetry-based representation by making a multivariate Gaussian approximation to the distribution, using the cross-covariance matrix of each batch to get the parameters of the relevant Gaussian. We use a low-rank approximation for the entropy and control the rank during the course of training. By initially starting from the lowest rank and increasing gradually to full rank, this method provides a favorable optimization path (see Appendix B). Considering the scaling properties of entropy with respect to rank, we use "per-rank entropy" via dividing estimated entropy to the rank of approximation.

## 5 Experiments

### 5.1 Setups

**One-parameter translation symmetry experiments** First, we evaluated our method on a synthetic 63-dimensional translation-invariant dataset. Each sample was generated by superposing multiple functions randomly selected from a family of smooth, compact signals $H := \{h_\theta\}$, where $\theta$ paramterizes the signal shape. We specifically employed two families for $H$: Gaussian functions and Associated Legendre Polynomials. This setting forms a preliminary test to method's ability for discovering symmetries when the dataset doesn't involve identical samples those are globally translated versions of each other. In other saying, we aim to see if the model can learn the symmetries when the group acts intransitively. For more details, see Appendix G.

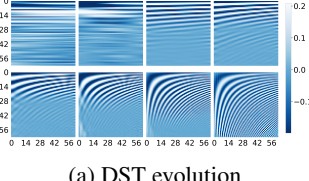

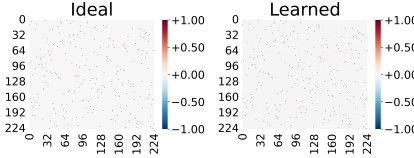

(a) DST evolution

(b) Symmetry generator for permuted ISING.

Figure 3: In Figure 3a, we show the training snapshots for a matrix that represents the lifting operation for frequency-shift symmetric distribution. Optimization follows a continuous trajectory, progressively building the inverse DST-I transformation. On the right, Figure 3b shows the inverse permutation matrix learned in the shuffled MNIST experiment.

**One-parameter frequency-shift symmetry experiments** To evaluate our model on a dataset with frequency shift symmetry, we applied DST-I transformation over the translation invariant dataset generated in Section 5.1. This allows us to test our method on symmetries that are less obvious to the human eye, thereby validating its bias-free nature.

**One-parameter permuted translation experiments** We tested our model over the synthetic dataset, generated by widely recognized ISING model. First we randomly assigned it spin configurations

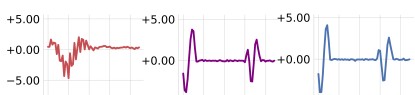 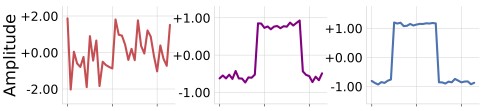

Figure 4: Waveform samples for one-parameter group experiments. At left we see transformed samples and the model undercovers the signal by learning the inverse transformation.

Table 1: Cosine similarities between the learned symmetry generators and the true, minimal generators.

| Signal | Input Dimensions | Batch Size | Epochs | Invariance | Generator A | Generator B | Input-Output Correlation |
|---|---|---|---|---|---|---|---|
| 1D Gaussian | 63 | 5000 | 10000 | Translation | 0.979 | – | 0.915 |
| 1D Legendre | 63 | 5000 | 10000 | Translation | 0.993 | – | 0.998 |
| 1D Gaussian | 63 | 5000 | 10000 | Frequency shift | 0.984 | – | 0.789 |
| 1D Legendre | 63 | 5000 | 10000 | Frequency shift | 0.988 | – | 0.979 |
| 1D Ising | 33 | 5000 | 7500 | Shuffled translation | – | 0.910 | 0.995 |
| 1D Ising | 33 | 5000 | 7500 | Linearly distorted translation | – | – | 0.992 |
| MNIST (15 x 15) | 225 | 5000 | 20000 | Translation | 0.961 | 0.963 | 0.995 |
| MNIST (15 x 15) | 225 | 5000 | 20000 | Shuffled translation | 0.976 | 0.970 | 0.994 |
| MNIST (27 x 27) | 729 | 5000 | 30000 | Shuffled translation | 0.675 | 0.722 | 0.710 |

to 33 dimensional lattice with entries in $\{-1, +1\}$. We draw the coupling strength $\beta$ uniformly for sample from a given range $[1.0, 5.0]$. The spins are then updated iteratively for a 10 Gibbs sweeps, where at each position the probability of flipping is determined by the local field (sum of neighboring spins) passed through a sigmoid function scaled by $\beta$. Afterwards, the resultant dataset is permuted by a random permutation matrix.

**One-parameter approximate translation experiments** For evaluating whether it's possible to recover more challenging non-orthogonal distortions, we corrupted the data with a systematic random linear transformation, whose singular values are limited to the range $[0.75, 1.33]$ for preventing information loss. Since the model is tailored for orthogonal group representations, we gave it a degree of freedom by plugged in a linear map $K : \mathbb{X} \to R^d$, before the lifting operation is applied.

**Two-parameter translation symmetry experiments** To test our method over two-parameter groups and it's scaling properties we used MNIST dataset. To enforce translation invariance, we first pad each image with zeros to a size of $84 \times 84$ pixels. We then generate two different datasets by randomly cropping with $15 \times 15$ and $27 \times 27$ patches, and flatten each patch into a 225 and 729-dimensional vectors, disrupting spatial structure.

**Two-parameter permutation symmetry experiments** As in Section 5.1 we ensure translation invariance by padding and randomly cropping. Then we apply a random permutation to each sample, converting them into bag of pixels. Finally, each patch is flattened into a 225 or 729-dimensional vector, requiring our model to autonomously reconstruct the spatial domain as a result of learning the transformed symmetry representation.

# 6 DISCUSSION

In this paper, we developed a symmetry and symmetry-based domain learning method for commutative Lie groups by using a representation learning perspective. We showed in variety of synthetic experiments that the developed method can learn symmetries on variety of experimental settings. Besides this, it can also invert mild distortions, recovering underlying domain. These problems have not been addressed previously in the unsupervised setting, by using symmetry, and we believe that this makes a unique contribution, opening new horizons.

However, currently the method is limited to abelian Lie groups, and it should be generalized to more general non-commutative symmetries to manifest it's full potential. Besides, our experiments show that datasets with Gaussian distributions are challenging for the current implementation. We have probed the origin of this problem as covariance based joint entropy approximation, which we plan to address in future studies.

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

## A PROOFS AND DERIVATIONS

### A.1 DECONVOLVING SYMMETRY-DOMAIN SIGNALS

Solving $f$ from $z$ is a deconvolution problem, whose solution we next outline in the case of a compact $G$, i.e., assuming $G$ is a $P$-dimensional torus $\mathbb{T}^P = \mathbb{S}^1 \times \ldots \times \mathbb{S}^1$, parametrized so that $t_j \equiv t_j + 2\pi$.

Let $\mathbf{u}_m$ be the eigenvectors of the Lie group representations. We will denote eigenvalues as a vector $\boldsymbol{\theta}_m := \left( \theta_m^{(0)}, \theta_m^{(1)}, ..., \theta_m^{(P)} \right)$. Then for any group element parametrized by $\mathbf{t}$, we can express its representation as $\rho(\mathbf{t}) = \sum_m \mathbf{u}_m \mathbf{u}_m^\dagger e^{i\boldsymbol{\theta}_m \cdot \mathbf{t}}$. The unitary irreducible representations (irreps) of an abelian Lie group are 1-dimensional and they can be labeled by grid of integers, with each corresponding to a P-dimensional irrep with $\mathbf{r}_m \in \mathbb{Z}$. We will denote $\boldsymbol{\theta}_m$ as $\mathbf{r}_m \in \mathbb{Z}^P$ similar to the one dimensional case.

The Fourier coefficients of a function $g(\mathbf{t})$ is given by $\tilde{G}_{\mathbf{n}} = \frac{1}{(2\pi)^P} \int_0^{2\pi} e^{-i\mathbf{n}\cdot\mathbf{t}} g(\mathbf{t}) d\mathbf{t}$, where $\mathbf{t}$ and $\mathbf{n}$ are vectors. While coefficients $\mathbf{n} \in \mathbb{Z}^P$ form a grid, $\mathbf{t} \in \mathbb{R}^P$ is a vector defined over the parameter space.

With this, we get,

$$\tilde{Z}_{\mathbf{n}} = \frac{1}{(2\pi)^P} \int_0^T \boldsymbol{\delta}_o^\top \rho(-\mathbf{t}) \mathbf{x}\, e^{-i\mathbf{n}\cdot\mathbf{t}}\, d\mathbf{t} \tag{13}$$

$$= \boldsymbol{\delta}_o^\top \left[ \sum_m \frac{1}{(2\pi)^P} \int_0^{2\pi} \mathbf{u}_m \mathbf{u}_m^\dagger\, e^{-i\mathbf{t}\cdot\mathbf{r}_m} e^{-i\mathbf{n}\cdot\mathbf{t}}\, d\mathbf{t} \right] \mathbf{x} \tag{14}$$

$$= \boldsymbol{\delta}_o^\top \left[ \sum_m \mathbf{u}_m \mathbf{u}_m^\dagger\, \delta_{\mathbf{n}+\mathbf{r}_m} \right] \mathbf{x} \tag{15}$$

Similarly,

$$\tilde{K}_n = \frac{1}{(2\pi)^P} \int_0^\top \boldsymbol{\delta}_{\boldsymbol{o}}^\top \rho(-\mathbf{t}) \boldsymbol{\delta}_{\boldsymbol{o}} e^{-i\mathbf{n}\cdot\mathbf{t}} d\mathbf{t} = \boldsymbol{\delta}_{\boldsymbol{o}}^\top \left[ \sum_m \mathbf{u}_m \mathbf{u}_m^\dagger \delta_{\mathbf{n}+\mathbf{r}_m} \right] \boldsymbol{\delta}_{\boldsymbol{o}}. \tag{16}$$

Then, the symmetry-based signal $f$ can be recovered by the inverse transform giving us the following expression;

$$f(\mathbf{t}) = \sum_m \frac{\boldsymbol{\delta_o}^\dagger \mathbf{u}_m \mathbf{u}_m^\dagger \mathbf{x}}{\boldsymbol{\delta_o}^\dagger \mathbf{u}_m \mathbf{u}_m^\dagger \boldsymbol{\delta_o}} \exp\left(-i\mathbf{t} \cdot \boldsymbol{r}_m\right) \tag{17}$$

$$= \left[\sum_l \frac{\boldsymbol{\delta_o} \mathbf{u}_l \mathbf{u}_l^\dagger}{\boldsymbol{\delta_o}^\dagger \mathbf{u}_l \mathbf{u}_l^\dagger \boldsymbol{\delta_o}}\right] \left[\sum_m \mathbf{u}_m \mathbf{u}_m^\dagger \exp\left(-i\mathbf{t} \cdot \boldsymbol{r}_m\right)\right] \mathbf{x} \tag{18}$$

$$= \boldsymbol{\phi}^\dagger \rho(-\mathbf{t})\mathbf{x}. \tag{19}$$

In the last equation, we defined the **resolution filter** $\boldsymbol{\phi} \in \mathbb{X}$ as

$$\boldsymbol{\phi}^\top := \sum_m \frac{\boldsymbol{\delta_o}^\dagger \mathbf{u}_m \mathbf{u}_m^\dagger}{\boldsymbol{\delta_o}^\dagger \mathbf{u}_m \mathbf{u}_m^\dagger \boldsymbol{\delta_o}} . \tag{20}$$

## A.2 CONVERTING GROUP INVARIANCE TO SHIFT INVARIANCE

The invariance of a random field under a transformation is defined in terms of the invariance of its finite-dimensional distributions. We restrict our attention to the case of continuous distributions. For a selection $\mathbf{t}_1, \ldots, \mathbf{t}_M$ of points on the group parameter space, the joint probability density of $(\hat{f}_1, \ldots, \hat{f}_n) := (\hat{f}(\mathbf{t}_1), \ldots, \hat{f}(\mathbf{t}_n))$. is given by

$$p_{\mathbf{t}_1, \ldots, \mathbf{t}_m}(\hat{f}_1, \ldots, \hat{f}_M) = \int_{\mathcal{X}} p_X(\mathbf{x}) \prod_{i=1}^M \delta\left(\hat{f}_i - \boldsymbol{\phi}^\top \rho(-\mathbf{t}_i)\mathbf{x}\right) d\mathbf{x}.$$

Consider a shift of the group parameter values $\mathbf{t}_j$ by a vector $\mathbf{a} \in \mathbb{R}^P$. The value of $p_M$ at the shifted values is given by

$$p_{\mathbf{t}_1+\mathbf{a}, \ldots, \mathbf{t}_m+\mathbf{a}}(\hat{f}_1, \ldots, \hat{f}_M) = \int p_X(\mathbf{x}) \prod_{i=1}^M \delta(\hat{f}_i - \boldsymbol{\phi}^\top \rho(-(\mathbf{t}_i + \mathbf{a}))\mathbf{x}) d\mathbf{x}$$

Applying the change of variables $\mathbf{x}' = \rho(-\mathbf{a})\mathbf{x}$ and noting that the orthogonality of the representation implies $d\mathbf{x}' = d\mathbf{x}$, we get:

$$p_{\mathbf{t}_1+\mathbf{a}, \ldots, \mathbf{t}_m+\mathbf{a}}(\hat{f}_1, \ldots, \hat{f}_M) = \int p_X(\rho(\mathbf{a}) \cdot \mathbf{x}') \prod_{i=1}^M \delta(\hat{f}_i - \boldsymbol{\phi}^\top \rho(-\mathbf{t}_i)\mathbf{x}') d\mathbf{x}'.$$

By the assumption of $G$-invariance, $p_X(\rho(\mathbf{a}) \cdot \mathbf{x}') = p_X(\mathbf{x}')$, therefore:

$$p_{\mathbf{t}_1+\mathbf{a}, \ldots, \mathbf{t}_m+\mathbf{a}}(\hat{f}_1, \ldots, \hat{f}_M) = \int p_X(\mathbf{x}') \prod_{i=1}^M \delta(z_i - \boldsymbol{\phi}^\top \rho(-\mathbf{t}_i)\mathbf{x}') d\mathbf{x}'$$

$$= p_{\mathbf{t}_1, \ldots, \mathbf{t}_m}(\hat{f}_1, \ldots, \hat{f}_M)$$

proving stationarity under translations.

### A.2.1 ON THE EFFECT OF RESOLUTION TERM

In this section, we will argue how resolution term favors the symmetry-based representation to resemble Markov field of lowest order. Currently, we will limit our setup for one-parameter groups for simplicity.

Let $\mathbf{Y}$ denote the $d$-dimensional output representation learned by the model. Using the chain rule for entropy, the joint entropy of $\mathbf{Y}$ can be written as $h(\mathbf{Y}) = \sum_{i=1}^d h(Y_{\sigma(i)}|Y_{\sigma(<i)})$ where $\sigma$ represents any permutation of the components. $Y_{\sigma(<i)}$ denotes the components $\{Y_{\sigma(1)}, Y_{\sigma(2)}, \ldots Y_{\sigma(i-1)}\}$ when $\sigma(i > 1)$, and for $Y_{\sigma(<1)}$, it simply means no conditioning is applied. We define the following $m$th order approximation to the chain rule formula that only uses $m$ components for conditioning:

$$h_\sigma^{(m)}(\mathbf{Y}) = \sum_i h(Y_{\sigma(i)}|Y_{\sigma([i-1, i-m])}) \tag{21}$$

where $Y_\sigma([i-1, i-m])$ denotes the random variables $\{Y_{\sigma(j)}\}_{i-m \le j \le m-1}$.

Since conditioning can only reduce entropy (Cover, 1999) we have

$$h(\mathbf{Y}) \le h_\sigma^{(d-1)}(\mathbf{Y}) \le \cdots \le h_\sigma^{(1)}(\mathbf{Y}) \le h_\sigma^{(0)}(\mathbf{Y}). \tag{22}$$

Using Equation 22, Total correlation $C(\mathbf{Y})$ can be written as sum of conditional mutual information terms;

$$C(\mathbf{Y}) = \sum_{m=0}^{d-1} h_\sigma^m(\mathbf{Y}) - h_\sigma^{m+1}(\mathbf{Y}) \tag{23}$$

$$= \sum_{m=0} \sum_{i=0} h(Y_{\sigma(i)}|Y_{\sigma([i-1,i-m])}) - h(Y_{\sigma(i)}|Y_{\sigma([i-1,i-m-1])}) \tag{24}$$

$$= \sum_{m=0} \sum_{i=0} I(Y_{\sigma(i)}; Y_{\sigma(i-m-1)}|Y_{\sigma([i-1,i-m])}) \tag{25}$$

$$= \sum_{i=0} I(Y_{\sigma(i)}; Y_{\sigma(i-1)}) + \sum_{m=1} \sum_{i=0} I(Y_{\sigma(i)}; Y_{\sigma(i-m-1)}|Y_{\sigma([i-1,i-m])}) \tag{26}$$

Hence giving us;

$$\sum_{m=1} \sum_{i=1} I(Y_{\sigma(i)}; Y_{\sigma(i-m-1)}|Y_{\sigma([i-1,i-m])}) = C(\mathbf{Y}) - \sum_{i=0} I(Y_{\sigma(i)}; Y_{\sigma(i-1)}) \tag{27}$$

However, lifting operation puts a lower bound to the covariance between any two components $Y_i$ and $Y_j$ of the symmetry-based representation, depending on the distance of two group elements in the parameter space. Taking $\mathbb{E}[Y_i] = 0$ for convenience, we can exchange the covariance $\text{Cov}(Y_{\sigma(i)}, Y_{\sigma(i-1)})$ between two components $Y_{\sigma(i)}$ and $Y_{\sigma(i-1)}$ by the average of squared distance $\overline{d^2}_{\sigma(i),\sigma(j)} := \mathbb{E}\left[d^2_{\sigma(i),\sigma(i-1)}\right] = \text{Var}(Y_{\sigma(i)}) + \text{Var}(Y_{\sigma(i-1)}) - 2\,\text{Cov}^2(Y_{\sigma(i)}, Y_{\sigma(i-1)})$. For convenience, we assume that $\text{Var}(Y_{\sigma(i)})$ is constant, which we denote by $\alpha \in \mathbb{R}$ and $\alpha > 0$. This assumption is also favored by the shift-invariance term. Then the average squared distance simplifies to $\overline{d^2}_{\sigma(i),\sigma(j)} = 2\alpha(1 - r_{\sigma(i),\sigma(i-1)})$.

Additionally, we know that distance between two components is upper bounded from the continuity argument 3.3. Then

$$\overline{d^2}_{\sigma(i),\sigma(i-1)} = 2\alpha(1 - r_{\sigma(i),\sigma(i-1)}) \le ||\phi||^2 |\rho(t_{\sigma(i)}) - \rho(t_{\sigma(i-1)})||_F^2 ||\mathbf{x}||^2$$

$$\implies r_{\sigma(i),\sigma(i-1)} \ge 1 - \frac{||\phi||\,||\mathbf{x}||}{2\alpha} ||\rho(t_{\sigma(i)}) - \rho(t_{\sigma(i-1)})||_F^2$$

For simplicity, lets define the distance $D_{ij} := \frac{||\phi||\,||\mathbf{x}||}{2\alpha}||\rho(t_i) - \rho(t_j)||_F^2$. Assuming that components $Y_i$ are Gaussian, lower bound to correlation introduces a lower bound to mutual information such that $I(Y_{\sigma(i)}; Y_{\sigma(i-1)}) \ge -\sum_i \frac{1}{2} \log(1 - r^2_{\sigma(i),\sigma(i-1)})$. Since we want to find the greatest lower bound for mutual information $I(Y_{\sigma(i)}; Y_{\sigma(i-1)})$, we will only consider the permutations such that $D_{\sigma(i),\sigma(j)} \le 1$. In this case we have

$$r^2_{\sigma(i),\sigma(i-1)} \ge \left[1 - D_{\sigma(i),\sigma(i-1)}\right]^2$$

$$\implies I(Y_{\sigma(i)}; Y_{\sigma(i-1)}) \ge -\sum_i \log(1 - \left[1 - D_{\sigma(i),\sigma(i-1)}\right]^2)$$

$$= -\sum_i \log((2 - D_{\sigma(i),\sigma(i-1)})D_{\sigma(i),\sigma(i-1)}).$$

This result shows that lower bound to mutual information $I(Y_{\sigma(i)}; Y_{\sigma(i-1)})$ becomes greatest when $D_{\sigma(i),\sigma(i-1)} \to 0$, hence $||\rho(t_i) - \rho(t_j)||_F^2 \to 0$ under the non-zero variance $\alpha$ assumption. Then, there exists a set of permutations $\{\sigma*_l\}$, which gives the highest lower bound to mutual information, and this will form continuous trajectories minimizing $\prod_i ||\rho(t_{\sigma*(i)}) - \rho(t_{\sigma*(i-1)})||_F$ among the

parameter space. For any of these permutations we can express the least upper bound for the sum of other conditional mutual information terms

$$\sum_{m=1}\sum_{i=1} I(Y_{\sigma*(i)}; Y_{\sigma*(i-m-1)}|Y_{\sigma*([i-1,i-m])}) \leq C(\mathbf{Y}) - I_{\sigma*} \tag{28}$$

where $I_{\sigma*} := -\sum_i \log((2 - D_{\sigma(i),\sigma(i-1)})D_{\sigma(i),\sigma(i-1)})$ is the lower bound for mutual information $I(Y_{\sigma(i)}; Y_{\sigma(i-1)})$.

Consequently, greatest upper bound for the sum of conditional mutual information terms is for the permutations $\sigma*$ that form continuous trajectories. Then, minimizing resolution favors Total correlation $C(\mathbf{Y})$ to be close to the tightest lower bound $I_{\sigma*}$. This means that conditional mutual information terms $\sum_{m=1}\sum_{i=1} I(Y_{\sigma*(i)}; Y_{\sigma*(i-m-1)}|Y_{\sigma*([i-1,i-m])})$ favors to be zero for this $\sigma*$.

For one-parameter case, which we limited our scope, trajectories can be guessed since there is only one continuous trajectory among the points of parameter space. This shows that for one-parameter groups, resolution loss term favors the symmetry-based representation to resemble a first order Markov field. Empirically, we observe that favors lowest order Markov field in general.

### A.3 Transformation of symmetry with orthogonal maps

For all $\mathbf{x} \in \mathbb{R}^d$ and $g \in G$ the invariance of the density $p_\mathcal{X}$ under the group action $\rho$ for $G$ is given as

$$p_\mathcal{X}(\mathbf{x}) = p_\mathcal{X}(\rho(g) \cdot \mathbf{x}) \tag{29}$$

The application of an orthogonal transformation $Q \in \mathbb{R}^{d \times d}$ to the data leads to a distribution which is invariant under the new group action $\rho'$

$$p(\mathbf{x}) = p(\rho(g) \cdot \mathbf{x}) \implies p(Q\mathbf{x}) = p(Q\rho(g) \cdot \mathbf{x}) \tag{30}$$

$$p(\mathbf{x}') = p(Q\rho(g) \cdot Q^{-1} \cdot \mathbf{x}') \implies \rho'(g) = Q\rho(g)Q^{-1} \tag{31}$$

where we defined $\mathbf{x}' = Q\mathbf{x}$.

### A.4 Transformation of origin with orthogonal maps

Based on the data model proposed in Section 3.2, applying an orthogonal transformation both changes the origin $\boldsymbol{\delta_o} \in \mathbb{X}$ as well as the symmetry representation.

To see, lets multiply both sides of the Equation 32 with $Q \in \mathbb{R}^d \times \mathbb{R}^d$

$$Q\mathbf{x} = \int_G f(v)Q\rho(v)\boldsymbol{\delta_o}d\mu(v) \tag{32}$$

$$= \int_G f(v)Q\rho(v)Q^\top Q\boldsymbol{\delta_o}d\mu(v) \tag{33}$$

$$= \int_G f(v)\tilde{\rho}(v)Q^\top \tilde{\boldsymbol{\delta_o}}d\mu(v) \tag{34}$$

where $\tilde{\rho}(v) := Q\rho(v)Q^\top$ and $\tilde{\boldsymbol{\delta_o}} := Q\boldsymbol{\delta_o}$.

This result also aligns with the Appendix A.3.

## B Training Details

We trained all models on NVIDIA A100 GPUs. Unless otherwise specified, we used the Adam optimizer with default hyperparameters (Table 3). The learning rate followed an exponential decay schedule, decreasing smoothly from the initial value to the final value listed in Table 2.

## C Implementation Details

### C.1 Efficient implementation for lifting

To compute (8) efficiently, we use the basis transformation $U$ that simultaneously diagonalizes $L_i$:

$$y_\mathbf{n} = \boldsymbol{\phi}^\top e^{-\sum_{i=1}^P t_{\mathbf{n}_i} L_i}\mathbf{x} = \boldsymbol{\phi}^\top U e^{-\sum_{i=1}^P t_{\mathbf{n}_i} D_i} U^\dagger \mathbf{x} = \tilde{\boldsymbol{\phi}}^\dagger e^{-\sum_{i=1}^P t_{\mathbf{n}_i} D_i} \tilde{\mathbf{x}} \tag{35}$$

Table 2: Experiment details.

| Experiment | Dataset size | Duration | Epochs | Eigendecomposition algorithm | Batch size | Primary initial lr | Auxiliary initial lr | $\frac{\text{Lr}_{\text{initial}}}{\text{Lr}_{\text{final}}}$ | Invariance weight | Resolution weight | Infomax weight | $\frac{\text{Noise(std)}}{\text{Signal(std)}}$ |
|---|---|---|---|---|---|---|---|---|---|---|---|---|
| Gaussian translation 63 timesteps | 500k | 11 hrs | 10000 | SVD | 5000 | $10^{-4}$ | $10^{-3}$ | 0.1 | 1.0 | 0.25 | 0.25 | 0.05 |
| Legendre translation 63 timesteps | 500k | 11 hrs | 10000 | SVD | 5000 | $10^{-4}$ | $10^{-3}$ | 0.1 | 1.0 | 0.25 | 0.25 | 0.05 |
| Gaussian frequency shift 63 timesteps | 500k | 11 hrs | 10000 | SVD | 5000 | $10^{-4}$ | $10^{-3}$ | 0.1 | 1.0 | 0.25 | 0.25 | 0.05 |
| Legendre frequency shift 63 timesteps | 500k | 11 hrs | 10000 | SVD | 5000 | $10^{-4}$ | $10^{-3}$ | 0.1 | 1.0 | 0.25 | 0.25 | 0.05 |
| MNIST 15x15 translation | 500k | 41 hrs | 20000 | SVD | 5000 | $10^{-4}$ | $10^{-3}$ | 0.1 | 1.0 | 0.2 | 0.2 | 0.00 |
| MNIST 15x15 permuted translation | 500k | 41 hrs | 20000 | SVD | 5000 | $10^{-4}$ | $10^{-3}$ | 0.1 | 1.0 | 0.2 | 0.2 | 0.00 |
| MNIST 27x27 permuted translation | 500k | 132 hrs | 30000 | EIG | 5000 | $10^{-4}$ | $10^{-3}$ | 0.1 | 1.0 | 0.10 | 0.10 | 0.00 |

Table 3: Optimizer hyperparameters.

| Parameter | Value |
|---|---|
| Beta 1 ($\beta_1$) | 0.9 |
| Beta 2 ($\beta_2$) | 0.999 |
| Epsilon ($\epsilon$) | $1 \times 10^{-7}$ |
| Weight Decay | 0 |
| Amsgrad | False |

where we defined $\tilde{\phi} := U^\dagger \phi$ and $\tilde{\mathbf{x}} := U^\dagger \mathbf{x}$ in the last expression, and $D_i \in \mathbb{C}^{d \times d}$ are the diagonal eigenvalue matrices for $L_i$. This implementation ensures both numerical stability and efficiency by exchanging the matrix exponential with scalar operations.

## C.2 LIE BASIS PARAMETRIZATIONS

For Abelian Lie groups, Lie basis elements must share eigenvectors. To ensure this we parametrize all eigenvectors by using a single anti-symmetric matrix. Eigenvalues for each Lie basis element are parametrized seperately as real vectors $\{\boldsymbol{\alpha}_i \in \mathbb{R}^d : i = 1, \ldots, P\}$.

To parametrize the Lie basis eigenvectors, we use a single learnable anti-symmetric matrix $A \in \mathbb{R}^{d \times d}$ and obtain the eigenvectors $U \in \mathbb{C}^{d \times d}$ by diagonalization. Eigenvalues $\{\phi_i\}$ for each basis are parametrized as seperate purely-imaginary vectors, and then exponentiated for obtaining eigenvalues of different group elements.

However, this parametrization require us to ensure that each Lie basis $L_i := U e^{(i\phi_i)}$, is an orthogonal matrix. This requires ensuring that seperatey parametrized eigenvalues and their corresponding eigenvectors form conjugate pairs.

To ensure this, we define a permutation matrix $P_{ij} := \mathbf{u}_i^\top \mathbf{u}_j$, which shuffles elements of any vector such that each component with index $j$ is mapped to the index satisfying $\mathbf{u}_i = \mathbf{u}_j^*$. Then we get properly ordered conjugate eigenvalue phases $\phi_i$ by applying the following operation

$$\phi_i = \boldsymbol{\alpha}_i - P\boldsymbol{\alpha}_i \tag{36}$$

where $\boldsymbol{\alpha}_i$ is the parameterization for the $i$'th Lie basis.

## C.3 COMPONENT-WISE ENTROPY ESTIMATION

We estimate component-wise entropies using trainable Gaussian mixture models for each component, following Pichler et al. (2022). This approach models the probability distribution $\hat{p}$ for each

component of y-representation as:

$$\hat{p}_{\mathbf{n}}(x) = \sum_{m=1}^{M} \frac{w_{\mathbf{n};m}}{\sigma_{\mathbf{n};m}\sqrt{2\pi}} \exp\left(-\frac{(x - \mu_{\mathbf{n};m})^2}{2\sigma_{\mathbf{n};m}^2}\right) \tag{37}$$

where $\mathbf{n} = (n_1, n_2, \ldots, n_k)$, with mixture weights satisfying $\sum_m w_{\mathbf{n};m} = 1$ and $w_{\mathbf{n};m} \geq 0$. We use $M = 4$ Gaussian components per mixture.

The component-wise entropy $h_{\mathbf{n}}$ is computed via:

$$h_{\mathbf{n}} = -\mathbb{E}_{y \sim P}\left[\log \hat{p}_{\mathbf{n}}(y_{\mathbf{n}})\right].$$

For training probability estimators, we minimize the average entropy $\frac{1}{d}\sum_{\mathbf{n}} h_{\mathbf{n}}$ across all components where $d$ denotes the total number of components.

### C.4 JOINT ENTROPY ESTIMATION

To estimate the multidimensional differential entropy of the symmetry-based representation $\mathbf{y}$, we use a multivariate Gaussian approximation. For a multivariate Gaussian with covariance matrix $\mathbf{C}$ whose eigenvalues are $\lambda_l$, the total entropy $h(\mathbf{y})$ is given, up to a constant shift, by $h(\mathbf{y}) = \sum_l h_l$ where $h_l = \log(\lambda_l)$.

For each batch, we flatten the symmetry-based representation and compute the sample covariance matrix $c_{pr} = \text{cov}(y_{\mathbf{i}(p)}, y_{\mathbf{j}(r)})$ where $p$ and $r$ correponds to flattened indexes. Then we apply eigendecomposition and sort its eigenvalues in descending order. Rank-$k$ approximation to the entropy as a weighted average of the per-component contributions to entropy is calculated as follows

$$\bar{h}_k \triangleq \frac{\sum_{l=1}^{d} w_{ki} \log(\lambda_l)}{\sum_{l=1}^{d} w_{kl}} \tag{38}$$

where the weights provide a soft thresholding at $l = k$.

We compute weights using the sigmoid function via

$$w_{kl} = \frac{1}{e^{\alpha(l-k)} + 1} \tag{39}$$

with $\alpha$ a hyperparameter determining the smoothness of the transition. Our experiments have shown consistent results for various values of $\alpha > 1$ (we chose $\alpha = 3.3$).

Due to the normalization, (38) should be thought of as a per-rank version of the low-rank entropy. In particular, when combining $\bar{h}_k$ with the marginal entropies of $y_{\mathbf{i}}$ during the computation of the total correlation loss of the Section 4.2.2, it is more appropriate to combine the former with the average marginal entropy rather than the total marginal entropy.

### C.5 JS-DIVERGENCE ESTIMATORS

#### C.5.1 OVERVIEW

In general, we probe shift-invariance by estimating the Jensen-Shannon (JS) divergence between the probability distribution of the $y$-representation and its shifted versions along each axis.

To estimate JS divergence, first we estimate 2 KL divergence term of each group axis, leveraging the dual representation which is given in Equation 12. Then use the KL divergences to compute JS divergence by a simple identity. This approach requires 2 networks for each group dimension. This approach enables us to formulate KL-divergence estimation as a highly efficient downstream task.

We use CNNs due to their stability and convergence speed, while we use position embeddings to improve their expressive capacity. For one-parameter 63 dimensional datasets, we also apply a coarse-grained symmetry-based representation, and estimate JS-divergence in two scales. Details to the coarse-graining procedure is given in Appendix C.5.3.

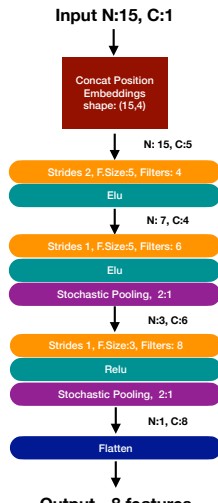

Figure 5: Estimator network figure toy model.

Figure 6: Dataset parameters

| Parameter | Value |
|---|---|
| Feature type | Gaussian |
| Sigma min | 0.5 |
| Sigma max | 1.5 |
| Amplitude Min | 0.5 |
| Amplitude Max | 1.5 |
| Noise std | 0.05 |
| Batch size | 2500 |

### C.5.2 POSITION EMBEDDINGS

The dual representation for KL divergence requires solving the optimization problem in Equation 12 over the space of all functions $f : \mathbb{R}^{d_1 \times d_2 \times \cdots \times d_k} \to \mathbb{R}$. In principle, deep multilayer perceptrons (MLPs) appear suitable for this task as they lack inductive biases. However, our experiments reveal that MLPs exhibit slow convergence, inducing instabilities in our method.

While Convolutional Neural Networks (CNNs) converge faster and more stably than MLPs, they can only represent translation-equivariant functions. To improve expressive capacity while preserving convergence efficiency, we introduce learnable position embeddings. The augmented y-representation becomes:

$$y^{\text{embed}}_{n_1,n_2,\ldots,n_k;e} := y_{n_1,n_2,\ldots,n_k} + p_{n_1,n_2,\ldots,n_k;e} \tag{40}$$

where $p \in \mathbb{R}^{d_1 \times d_2 \times \cdots \times d_k \times d_e}$ denotes the learnable position embedding tensor, and $d_e = 4$ is the embedding dimension used throughout our experiments.

### C.5.3 COARSE-GRAINING

Experiments indicate that high dimensionality along an axis can cause optimization challenges, such as optimizers becoming trapped in local minima. To mitigate this issue, we introduce a coarse-grained version of the $y$-representation and employ an additional JS-divergence estimator, running in lower dimensional symmetry-based representation.

The coarse-graining procedure consists of two steps:

1. Partition the $y$-representation into patches along each axis
2. Randomly select one component within each patch

This operation is formalized in Equation 41, where the coarse-grained representation $y^{\text{coarse}}$ is indexed by positive integers $(n_1, n_2, \ldots, n_k)$. For each sample, we independently generate random integer offsets $\{r_i\}$ uniformly distributed in $\{0, 1, \ldots, q_i - 1\}$, where $q_i$ denotes the dilation factor along axis $i$ for $i = 1, \ldots, k$. The indices $(n_1, n_2, \ldots, n_k)$ range over $(1, 2, \ldots, \lfloor d_1/q_1 \rfloor) \times \cdots \times (1, 2, \ldots, \lfloor d_k/q_k \rfloor)$.

$$y^{\text{coarse}}_{n_1,n_2,\ldots,n_k} := y_{n_1 q_1 + r_1, n_2 q_2 + r_2, \ldots, n_k q_k + r_k} \tag{41}$$

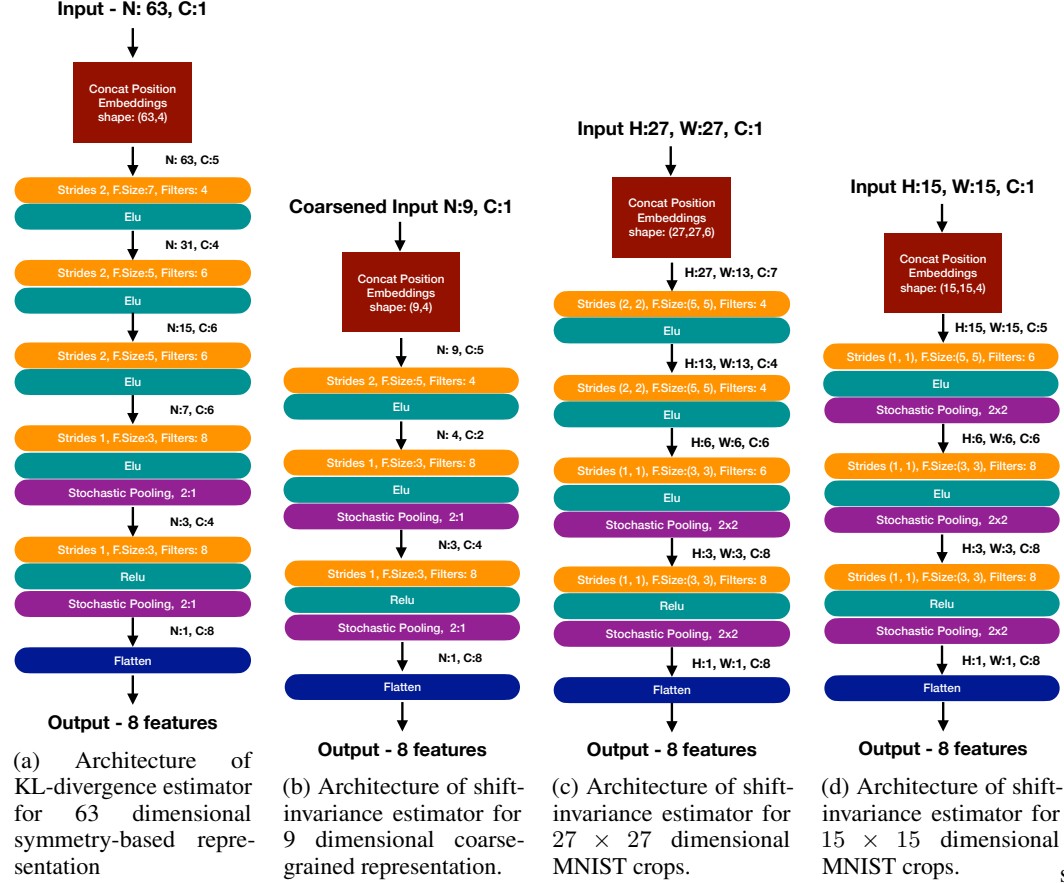

(a) Architecture of KL-divergence estimator for 63 dimensional symmetry-based representation

(b) Architecture of shift-invariance estimator for 9 dimensional coarse-grained representation.

(c) Architecture of shift-invariance estimator for $27 \times 27$ dimensional MNIST crops.

(d) Architecture of shift-invariance estimator for $15 \times 15$ dimensional MNIST crops.

### C.5.4 ARCHITECTURES OF KL-DIVERGENCE ESTIMATORS

Our networks are standard CNN downstream networks, with the additional position embedding at input. At each layer, we either use strides or pooling layer to downsample the features along spatial axis. Additionally, we constrain the spectral norm of each network to $10.$, by applying a layerwise spectral norm limitation which translates to $10^{\frac{1}{\#\text{layers}}}$ for each layer. This is required since exponentiation operation given in the dual representation leads to saturated estimations otherwise.

To reduce the computational costs, we use strides in the first few layer of the each network, since these are the most compute intensive parts. In the remaining layers we use stochastic pooling Zeiler & Fergus (2013) to battle with the curse of dimensionality.

In two-parameter experiments, we use 2 networks for each group dimension, making 4 identical networks in total. For one-parameter 63 dimensional experiments we form an additional coarse-grained representation, and quantify shift-invariance at two levels. To obtain a coarse-grained representation, we form patches of 7 components, and then randomly pool single component from each patch leading to 9 component coarse-grained representation.

In Figure 7a we see the structure of KL-divergence estimator network which runs over the whole symmetry-based representation. Figure 7b KL-divergence estimator for the coarse-grained representation. Due to lower dimensions, estimator running over the coarse-grained representation has very little computational costs.

In Figure 7d and Figure 7d, we see the structures of KL-divergence estimators for 15x15 and 27x27 dimensional MNIST datasets. In both cases, we use 4 identical networks, 2 networks for each axis.

# D ANALYZING THE EFFECT OF HYPERPARAMETERS

## D.1 ABLATION STUDIES

We conducted ablation studies over 1D, 63 dimensional datasets which are composed of Gaussian features. Results imply that all terms togather learn the symmetry and symmetry-based representation. Learned generators can be found in Figures 8.

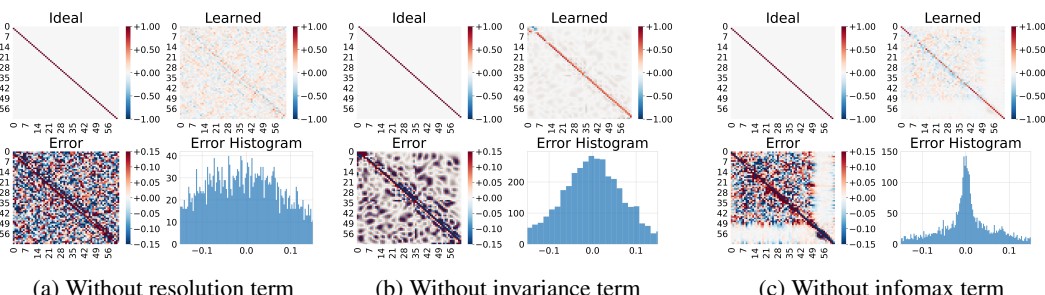

(a) Without resolution term     (b) Without invariance term     (c) Without infomax term

Figure 8: Ablation experiments conducted over 1D 63 dimensional Gaussian dataset. We see that training fails if any of the loss terms are dropped. However, it's interesting to see that without shift-invariance term, model is able to recover a noisy version of symmetry generator. To our interpretation, this implies the strength of locality prior.

## D.2 SENSITIVITY ANALYSIS

We applied Morris Sensitivity analysis over a toy model working trained over 15 dimensional dataset formed from Gaussian basis signals. We run 32 experiments and varied estimator learning rate, model learning rate, learning rate decay, shift-invariance, resolution and information maximization hyperparameters. Estimator network architecture and dataset properties can be found in Figure 6 and Table 5.

The result of sensitivity analysis implies that model learning rate and resolution term has more dominant importance compared to other terms. Other loss weights are almost insignificant. This finding also aligns with the relative ease of hyperparameter tuning. See Table 5 for the experiment parameters and their performance evaluation (cosine similarities).

Table 4: Morris sensitivity analysis results

| Parameter | $\mu$ | $\mu*$ | $\sigma$ | $\mu*$ **confidence** |
|---|---|---|---|---|
| estimator_lr | -0.048042 | 0.055294 | 0.065549 | 0.045945 |
| model_lr | 0.301463 | 0.301463 | 0.171484 | 0.133418 |
| lr_decay | 0.017997 | 0.068293 | 0.094078 | 0.044539 |
| uniformity | -0.044748 | 0.089709 | 0.104208 | 0.043226 |
| resolution | 0.122241 | 0.122241 | 0.067775 | 0.061329 |
| infomax | -0.004228 | 0.025048 | 0.033888 | 0.015398 |

# E COMPUTATIONAL COMPLEXITY

Our model has polynomial computation complexity of $\frac{d^3}{B}$ due to diagonalization, where $d$ is the dimensionality of input and $B$ is batch size. However, since they are applied in a per-batch manner, we didn't encounter any hindering effect till MNIST 27x27 dataset.

For MNIST 27x27, we reduced the computational cost by exchanging SVD with eigendecomposition algorithms. However, this leads to compromising accuracy, possibly due to SVDs higher stability.

Except the per-batch diagonalization algorithms, other components of our model is scalable to higher dimensions. Table 6 summarizes the complexities.

Table 5: Sensitivity experiment parameters and experiment results.

| Estimator LR $(\times 10^{-3})$ | Model LR $(\times 10^{-3})$ | $LR_{\text{initial}}/$ $LR_{\text{final}}$ | Invariance weight | Resolution weight | InfoMax weight | Cosine similarity |
|---|---|---|---|---|---|---|
| 0.0008 | 0.0001 | 0.1071 | 0.7857 | 0.0500 | 0.1357 | 0.7167 |
| 0.0008 | 0.0001 | 0.1071 | 0.7857 | 0.0500 | 0.1357 | 0.9802 |
| 0.0008 | 0.0001 | 0.1071 | 1.3571 | 0.0500 | 0.1357 | 0.8758 |
| 0.0008 | 0.0001 | 0.1071 | 1.3571 | 0.0500 | 0.1357 | 0.8932 |
| 0.0008 | 0.0001 | 0.1071 | 1.3571 | 0.1071 | 0.1357 | 0.9811 |
| 0.0014 | 0.0001 | 0.1071 | 1.3571 | 0.1071 | 0.1357 | 0.9899 |
| 0.0014 | 0.0001 | 0.1071 | 1.3571 | 0.1071 | 0.0786 | 0.9638 |
| 0.0014 | 0.0001 | 0.0500 | 1.3571 | 0.1071 | 0.0786 | 0.9609 |
| 0.0006 | 0.0001 | 0.1071 | 1.3571 | 0.0786 | 0.1357 | 0.7789 |
| 0.0006 | 0.0001 | 0.1071 | 1.3571 | 0.1357 | 0.1357 | 0.7947 |
| 0.0006 | 0.0001 | 0.1071 | 1.3571 | 0.1357 | 0.1357 | 0.8055 |
| 0.0012 | 0.0001 | 0.1071 | 1.3571 | 0.1357 | 0.1357 | 0.7252 |
| 0.0012 | 0.0001 | 0.0500 | 1.3571 | 0.1357 | 0.1357 | 0.6410 |
| 0.0012 | 0.0001 | 0.0500 | 1.3571 | 0.1357 | 0.0786 | 0.6682 |
| 0.0012 | 0.0001 | 0.0500 | 1.3571 | 0.1357 | 0.0786 | 0.9420 |
| 0.0012 | 0.0001 | 0.0500 | 0.7857 | 0.1357 | 0.0786 | 0.9986 |
| 0.0008 | 0.0001 | 0.0500 | 0.9286 | 0.1500 | 0.0643 | 0.9957 |
| 0.0008 | 0.0001 | 0.0500 | 0.9286 | 0.1500 | 0.1214 | 0.9838 |
| 0.0008 | 0.0001 | 0.0500 | 0.9286 | 0.0929 | 0.1214 | 0.8701 |
| 0.0008 | 0.0001 | 0.0500 | 1.5000 | 0.0929 | 0.1214 | 0.8519 |
| 0.0008 | 0.0001 | 0.0500 | 1.5000 | 0.0929 | 0.1214 | 0.6165 |
| 0.0008 | 0.0001 | 0.1071 | 1.5000 | 0.0929 | 0.1214 | 0.5494 |
| 0.0014 | 0.0001 | 0.1071 | 1.5000 | 0.0929 | 0.1214 | 0.5503 |
| 0.0014 | 0.0001 | 0.1071 | 1.5000 | 0.0929 | 0.1214 | 0.5551 |
| 0.0008 | 0.0001 | 0.1214 | 0.6429 | 0.0643 | 0.0929 | 0.8883 |
| 0.0008 | 0.0001 | 0.1214 | 0.6429 | 0.1214 | 0.0929 | 0.9968 |
| 0.0008 | 0.0001 | 0.1214 | 0.6429 | 0.1214 | 0.1500 | 0.9985 |
| 0.0008 | 0.0001 | 0.1214 | 0.6429 | 0.1214 | 0.1500 | 0.9263 |
| 0.0008 | 0.0001 | 0.1214 | 0.6429 | 0.1214 | 0.1500 | 0.8951 |
| 0.0008 | 0.0001 | 0.0643 | 0.6429 | 0.1214 | 0.1500 | 0.8671 |
| 0.0014 | 0.0001 | 0.0643 | 0.6429 | 0.1214 | 0.1500 | 0.8096 |
| 0.0014 | 0.0001 | 0.0643 | 1.2143 | 0.1214 | 0.1500 | 0.8696 |

Table 6: Computational complexity

| Component entropy estimators | KL-divergence estimators | Joint entropy estimator | Lifting |
|---|---|---|---|
| $O(d)$ | $O(d)$ | $O(d^2 + \frac{d^3}{B})$ | $O(d^2 + \frac{d^3}{B})$ |

# F  CAVEATS

## F.1  WEIGHTING JS-DIVERGENCE ESTIMATIONS

### F.1.1  SCALING PROPERTIES OF JS-DIVERGENCE

In our method, we quantify shift-invariance by estimating the Jensen-Shannon (JS) divergence between the probability distribution $P$ of the y-representation and its shifted variants along each axis. However, when axes have substantially different dimensionalities as a result of coarse graining, averaging JS divergences without accounting for dimensional effects could degrade performance.

To address this, we weight JS divergence estimates based on their scaling behavior with respect to dimensionality. We assume the y-representation is the discrete version of a rectangular region of a Lie manifold which is reflected in data distribution. Under this assumption, increasing the dimensionality along an axis corresponds to nothing more than increasing resolution, allowing us to estimate the scaling behavior of JS divergence.

Formally, we can parametrize $P$ with real parameters $\theta$ such that a small shift in y-representation corresponds to a perturbation $\delta\theta$ in parameter space. Expanding the JS divergence in a Taylor series

to second order at $\theta_0$ we have

$$\text{JS}\big(P(\theta_0) \parallel P(\theta_0 + \delta\theta)\big) = \text{JS}\big(P(\theta_0) \parallel P(\theta_0)\big) \tag{42}$$

$$+ \sum_i \frac{\partial}{\partial\theta_i}\text{JS}\big(P(\theta_0) \parallel P(\theta)\big)\Big|_{\theta=\theta_0}\delta\theta_i \tag{43}$$

$$+ \frac{1}{2}\sum_{i,j} \frac{\partial^2}{\partial\theta_i\partial\theta_j}\text{JS}\big(P(\theta_0) \parallel P(\theta)\big)\Big|_{\theta=\theta_0}\delta\theta_i\delta\theta_j \tag{44}$$

$$+ \mathcal{O}(\|\delta\theta\|^3) \tag{45}$$

. Since $\text{JS}\big(P(\theta_0) \parallel P(\theta_0)\big) = 0$ and the first derivative vanishes at the global minimum $\theta = \theta_0$, we obtain the second-order approximation:

$$\text{JS}\big(P(\theta_0) \parallel P(\theta_0 + \delta\theta)\big) \approx \frac{1}{2}\sum_{i,j} \frac{\partial^2}{\partial\theta_i\partial\theta_j}\text{JS}\big(P(\theta_0) \parallel P(\theta)\big)\Big|_{\theta=\theta_0}\delta\theta_i\delta\theta_j \tag{46}$$

Under Equation 46, scaling the shift magnitude by $s > 0$ (which scales $\delta\theta \to s\delta\theta$ for small continuous shifts) yields:

$$\text{JS}\big(P(\theta_0) \parallel P(\theta_0 + s\delta\theta)\big) \approx s^2 \cdot \text{JS}\big(P(\theta_0) \parallel P(\theta_0 + \delta\theta)\big), \tag{47}$$

demonstrating quadratic scaling of JS divergence with shift magnitude.

Since we estimate $\text{JS}(P \parallel P_l)$ for each axis $l$ where $P_l$ is the one-component shifted version of $P$ along $l$'th axes, and we assume that increasing dimensionality merely changes the resolution, magnitude of one-component shift should be proportional to $1/d_l$. Then from Equation 47, we conclude that $\text{JS}(P \parallel P_l) \propto 1/d_l^2$ while everything else kept the same.

To maintain consistency with the use of per-dimension entropy in Total Correlation and Infomax losses, we further divide by JS-divergences by the total dimension $d := \prod_{i=1}^{k} d_i$. This weighting is on the same footing with other loss term which is justified by considering the mutual information form of JS divergence

$$\text{JS}(P \parallel P_l) = h(P) - h\left(P \,\Big|\, \frac{P + P_l}{2}\right) \tag{48}$$

where the per-dimension expression $\frac{1}{d}\text{JS}(P \parallel P_l) = \frac{1}{d}h(P) - \frac{1}{d}h\left(P \,\big|\, \frac{P+P_l}{2}\right)$ includes joint and conditional entropies per dimension.

Thus, we assign the weight $w_l := d_l^2/d$ to the JS divergence along axis $l$. For coarse-grained representations, we exchange $d_l$ with $d_l/q_l$ and $d$ with $\prod_{i=1}^{k} d_i/q_i$ which is actually dimensions after the coarse-graining took place. This scheme ensures consistent hyperparameter settings across experiments when the stated assumptions hold.

Consequently, we rescale the JS-divergence terms $D_{\text{JS}}(P_{\mathbf{Y}}\|P_{\mathbf{Y}^{(l)}}) \to w_l D_{\text{JS}}(P_{\mathbf{Y}}\|P_{\mathbf{Y}^{(l)}})$.

### F.2 CONTROLLING THE RANK OF THE JOINT ENTROPY ESTIMATION

To help facilitate efficient optimization by first fitting to the gross features of the data, and refining over time, we use a time-dependent rank parameter $k$ for the entropy estimator (39). To adjust $k$ during training, we use a normalized notion of training time $t_n$ measuring the "amount of gradient flow" via

$$t_n = \frac{\sum_{s=1}^{n} \text{lr}(s)}{\sum_{s=1}^{T} \text{lr}(s)} \tag{49}$$

where lr(s) is the learning rate used at the training step (batch) $s$, and $n$ and $T$ are the current training steep, and the total number of training steps, respectively. We control the rank $k$ of the low-rank entropy estimator by setting $k = \text{ceil}(d \times t_n)$ so that by the end of the training, the rank is at $d$.

### F.3 PADDING

We use padding for the symmetry generator and the filter in the sense that the symmetry matrix and the filter have dimensionality that is higher than the dimensionality of the data, but we centrally crop the matrix and the filter before applying them to the data. This is done to deal with finite size (edge) effects, and after experimenting with padding sizes of 6 to 63, we saw that the results are not sensitive to padding size. Working with a cyclic/periodic symmetry would make the padding unnecessary, but this would mean working with a restrictive assumption on the underlying symmetry.

### F.4 INITIALIZATION OF EIGENVALUE PHASES

We initialize the norm of Lie basis eigenvalues using a normal distribution with $\sigma = 10^{-3}$ and $\mu = 0$. Using smaller standard deviations did not affect performance, however, significantly larger $\sigma$ values may lead to corrupt the optimization trajectory.

### F.5 NOISE INJECTION TO THE RESOLUTION FILTER

We initialize the resolving filter with zeros and add Gaussian noise during the early stages of training before computing the loss for each batch.

$$\psi \leftarrow \psi + \mathcal{N}(\mu = 0, \sigma = 0.1)\exp(-(t/\tau)) \tag{50}$$

The amplitude of the noise is set to decay exponentially with a short time constant (of $\tau = 10$ epochs). As mentioned previously, to compute the transformed data $\mathbf{y}$, we use a normalized version of $\theta$ at each step: $\hat{\psi} = \psi/\|\psi\|_2$

## G FURTHER RESULTS

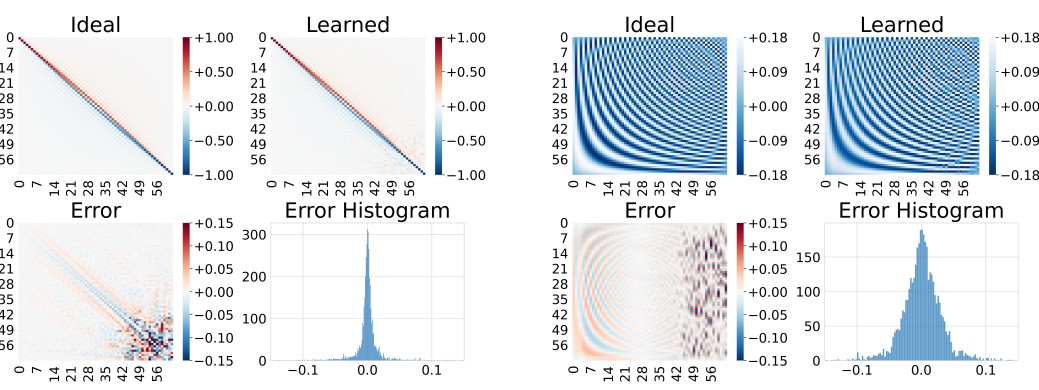

(a) Ideal and learned symmetry generators      (b) Ideal and learned group convolution tensors

Figure 9: Training results for the 1D frequency-shift-invariant dataset obtained via randomly placing Associated Legendre Polynomials in frequency space. On the left, we see the minimal generator of the frequency shift transformation, which transforms a sinusoidal basis signal to the next one according to the order in frequency. On the right, we see the group convolution matrix formed by repeatedly applying the frequency-shift generator to the resolving filter. This matrix is the (transpose of the) DST-I transformation matrix. In order to recover the symmetry-based local representation, the model learned to negate the transformation, recovering the domain where the signals are transparently local and symmetric.

In the 1D case, the ideal symmetry generator for the translation-invariant dataset is the 1-step translation operator, which is simply a shift matrix, with entries just below (or above) the diagonal equaling 1, all other entries being zero. In Figure 10a, we see that this matrix is learned to a high degree of accuracy except in some regions. This problem is due to the decimation procedure developed for computational efficiency. The group convolution matrix that gives the symmetry-based representation for translation symmetry is simply the identity operator. In Figure 10b, we see that this

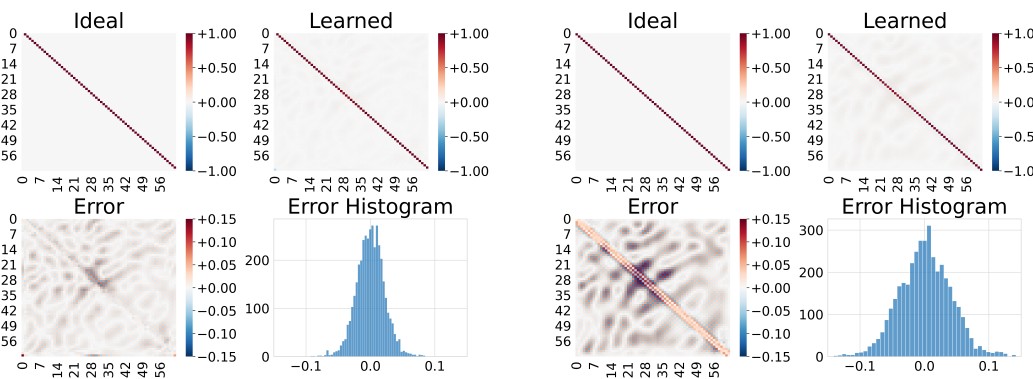

(a) Ideal and learned symmetry generators (top) and error distributions (bottom).

(b) Ideal and learned group convolution matrices (top) and error distributions (bottom).

Figure 10: Results for the 1D translation invariant data distribution obtained from Associated Legendre polynomials.

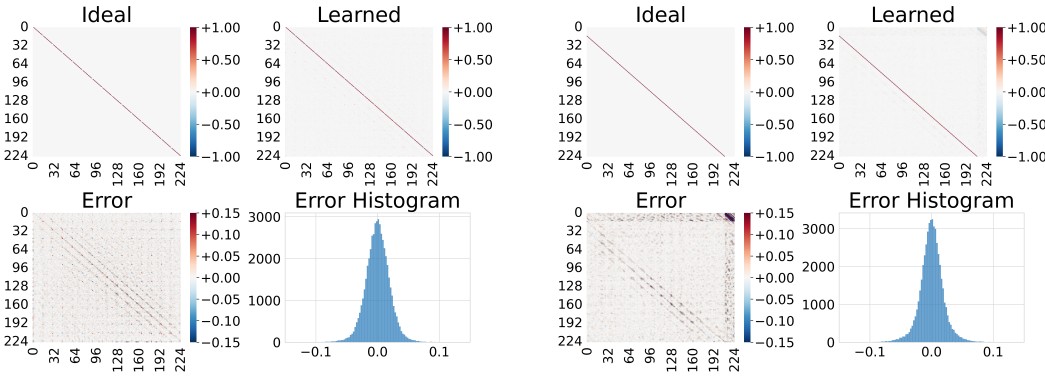

(a) Ideal and learned symmetry generators (top) and error distributions (bottom).

(b) Ideal and learned group convolution matrices (top) and error distributions (bottom).

Figure 11: The symmetry generators for the cropped MNIST dataset with permuted pixels. The dataset is invariant and local under the action of the powers of this permuted x-axis and y-axis translations.

operator, which is formed by combining the powers of the group generator with the learned resolving filter, is also learned with decent accuracy. However, there is a discontinuity in the intermediate segment, which is also due to the decimation procedure.

For the 2D case with a permuted-translation symmetry (obtained by applying a fixed permutation to the flattened, cropped MNIST samples), we see that there aren't any dominating flaws and errors are more homogenously distributed 11. We see the two complementary generators are indeed learned.

# H DATASETS

## H.1 1D DATASETS

**Data Generation Procedure** The synthetic samples were generated through the following procedure:

- For each sample $\mathbf{x}$, first determine the number of constituent signals by drawing $P \sim \mathcal{U}\{0, 1, \ldots, P_{\max}\}$ with $P_{\max} = 10$.
- For each signal $n = 1$ to $P$, draw shape parameters $\theta_n \sim \mathcal{U}(\Theta)$ and draw a translation $\tau_n \sim \mathcal{U}[-L, L]$.

- Construct the discrete-time signal via superposition: $\mathbf{x}[i] = \sum_{n=1}^{P} h_{\theta_n}(t_i - \tau_n)$ for $i = 1, \ldots, 63$, where $t_i$ defines a discrete time grid restricted between $[-L/2, L/2]$ to prevent boundary effects distorting translation invariance.

- Finally, we add Gaussian noise to the sample (with $\sigma = 0.05$).

This procedure can give us any symmetry that is related to component translations via a similarity transformation. See Figures 12a and 12b, which involve samples from datasets with different kinds of symmetries.

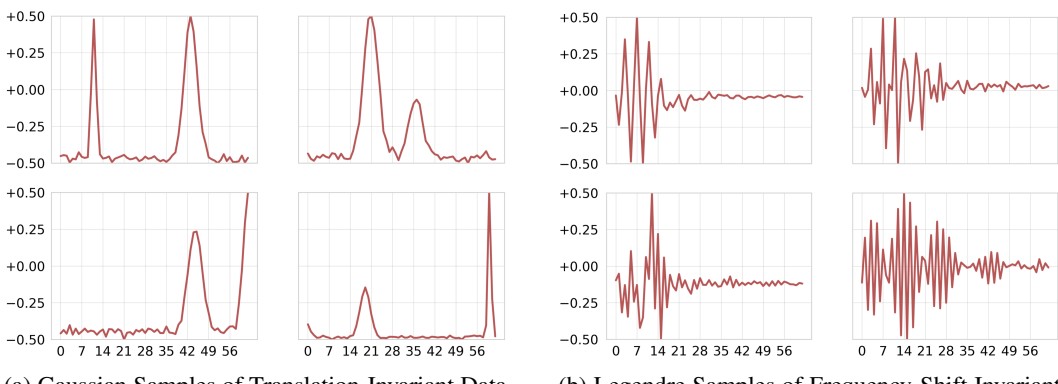

(a) Gaussian Samples of Translation-Invariant Data Distribution

(b) Legendre Samples of Frequency-Shift Invariant Data Distribution

Figure 12: Invariant Data Distributions: Gaussian vs. Legendre

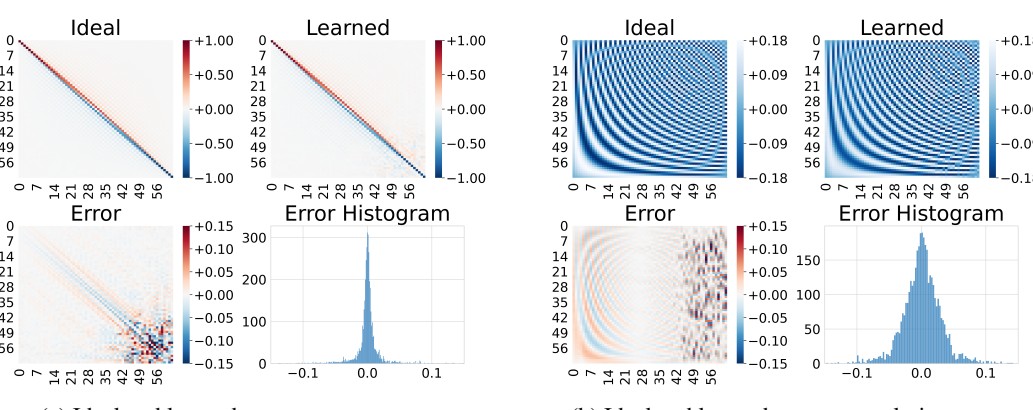

(a) Ideal and learned symmetry generators

(b) Ideal and learned group convolution tensors

Figure 13: Training results for the 1D frequency-shift-invariant dataset obtained via randomly placing Associated Legendre Polynomials in frequency space. On the left, we see the minimal generator of the frequency shift transformation, which transforms a sinusoidal basis signal to the next one according to the order in frequency. On the right, we see the group convolution matrix formed by repeatedly applying the frequency-shift generator to the resolving filter. This matrix is the (transpose of the) DST-I transformation matrix. In order to recover the symmetry-based local representation, the model learned to negate the transformation, recovering the domain where the signals are transparently local and symmetric.

**Basis Signal Types Gaussian signals** $f_{gaussian}(z; \mathcal{A}, \mu, \sigma)$ are parametrized by amplitude $\mathcal{A}$, center $\mu$, and width $\sigma$. The input $z$ is an integer ranging from $-32$ to $32$, labeling the components of the raw sample vectors. We sample the center $\mu$ uniformly from the extended (tripled) range $-97$ to $97$, and then crop the resulting signals to the $z$ range of $-32$ to $32$ to allow for the possibility of signals that contain only a tail of a Gaussian. $A$ and $\sigma$ are sampled from the ranges given in Table 7.

**Legendre signals** are given in terms of the associated Legendre polynomials and give localized waveforms that can change signs. The relevant parameters are center $c$, scale $s$, amplitude $\mathcal{A}$, and

the orders $l$, $m$: $f_{legendre}^{(l,m)}(z; \mathcal{A}, c, s) = \mathcal{A} P_l^m \left( \cos \left( \frac{z-c}{s} \right) \right)$. We crop these signals to the range $|x - c|/s \leq \pi$, i.e., set the values outside this range to zero.

Once again, $z$ becomes the discrete dimension index, ranging from $-32$ to $32$. For the $l, m$ parameters, we use $l = 2, m = 1$ and $l = 3, m = 1$, with equal probability for each sample. We sample the centers as in the Gaussian case, and the sampling of the other parameters is described in Table 7.

Table 7: The synthetic datasets prepared for experiments. The parameters for each basis signal are sampled from a uniform distribution with the indicated ranges.

| Signal Type | Input dimensions | Transform | Amplitude range | Scale range |
|---|---|---|---|---|
| Gaussian | 63 | Identity | [0.5, 1.5) | [0.5, 2.5) |
| Legendre (l=2-3, m=1) | 63 | DST-I | [0.5, 1.5) | [6.0, 25.0) |

## H.2  2D DATASETS

We obtain 2D datasets by zero-padding the MNIST dataset within each axis on two sides with $28$ pixels, leading to an image with size $84 \times 84$. Then, we crop this image by using a random window to $15 \times 15$ or $27 \times 27$ patches, ensuring uniform sampling along translations. Since this introduces too many blank samples (where all elements are zero), we drop images according to a maximum element via simple thresholding ($10^{-7}$) for efficient training. However, the model works without dropping the blank samples as well.

After cropping, we obtain versions whose distribution is invariant under different operations: translation and permuted translation. We obtain the latter by randomly shuffling pixels of the image, completely destroying neighborhood information.

