# OpenReview forum: "Unsupervised discovery of symmetries and symmetry-based domains from raw data"
_ICLR.cc/2026/Conference — Submitted to ICLR 2026_

### Official Review · Reviewer_SCTN · 2025-10-21

**Soundness:** 2
**Presentation:** 1
**Contribution:** 2
**Rating:** 2
**Confidence:** 3

**Summary:**

The paper proposes an unsupervised framework to jointly discover symmetries and the underlying symmetry-based data domain. It introduces a lifting operation that maps observed samples into a group-parameterized domain where symmetry becomes equivalent to shift-invariance. Then, the method combines invariance, resolution, and infomax losses to ensure nontrivial, independent, and informative representations and parameterizes group generators through learnable Lie algebra bases.

**Strengths:**

The paper tackles the ambitious and important problem of unsupervised discovery of symmetry structures directly from raw data. It uses a lifting-based approach that connects group invariance to shift stationarity. The formulation is mathematically grounded and the methodology is described in clear details.

**Weaknesses:**

## Presentation

The experiment section of this paper is **incomplete**. In Sec 5, there is only one subsection describing the experiment setups. Figure 3 and 4 and Table 1 may have presented some results, are not referenced anywhere in the text. Some other results are provided in the Appendix. However, a paper should be self-contained without the Appendix. It is okay that some complementary results and detailed dataset setups, for example, are deferred to the Appendix. However, in its current state, the paper does not effectively show and explain any experimental results in the main text, which is not acceptable.

In other sections beyond experiments, the paper also relies heavily on cross-references to the Appendix. Sometimes it explicitly links equations from the appendix, e.g., L231, (20). Sometimes notations from the appendix are directly used in the main text without reference or definition, which is even worse, e.g., L227, $\phi$. The mathematical framework introduced in the methodology may be correct, but the reliance on back-and-forth cross-references and intertwined notational dependency make it less accessible to readers.

## Method & Contribution

First and foremost, it appears to me that the contributions mentioned at the end of Sec 1 do not match the methodology in Sec 4. Either the summarized contribution points are inaccurate, or the methodology section has not provided a clear enough explanation. For example, one of the contributions is summarized as the "ability to discover the symmetries when they act intransitively", but intransitivity of group actions is not mentioned anywhere in the methodology. Similarly, another contribution point states that the method discovers irreducible representations, which are again not discussed anywhere else in the main paper. These go beyond just presentational issues, since they confuse the readers about the core contribution of the paper and make it difficult to evaluate the correctness and the significance of the proposed methodology.

Then, from my understanding, the paper aims to discover a hidden space where the symmetry of the data distribution acts simply with a regular representation, i.e., shift invariance (wrt an Abelian group). The paper presents a convoluted way to achieve this, while I believe there are simpler solutions. For example, [1,2,3], among other methods for learning equivariant representations, can all be applied once we specify the group action on the hidden space (i.e., the symmetry-based domain in the context of this paper). Also, compared to the setting in this paper, these existing methods are not limited to (1) Abelian groups, and (2) data domains exactly being the group $\Omega=G$, but instead more general spaces equipped with a $G$-action. Therefore, a critical question is why the proposed method in this paper is superior to existing works in terms of applicability, efficiency, etc.

## Experiments

As mentioned above, the current experiment section is incomplete. Thus, it is hard to provide any meaningful evaluation for this part.

## References

[1] Latent Space Symmetry Discovery. ICML 2023.

[2] Structuring representations using group invariants. NeurIPS 2022.

[3] Neural Fourier Transform: A General Approach to Equivariant Representation Learning. arxiv 2023.

**Questions:**

* L317: Missing cross-reference.
* L342: How is the JS divergence in the invariance loss estimated? It is better to explicitly refer to the relevant section instead of saying "... is described below", which is confusing because it is not described *immediately* below.
* L349: There are $N_1 \times ... \times N_P$ different $\mathbf n$'s, scaling quickly with the resolution and the group dimension. Is the summation over all possible $\mathbf n$'s efficient?

---

> ### Author Response · Authors · 2025-12-03
>
> Dear reviewer,
>
> Thank you for your valuable feedback. We have decided to reorganize our manuscript for another conference.
>
> To address your questions:
>
> **L317: Missing cross-reference**
>
> We apologize for the inconvenience. We will improve our presentation.
>
> **L342: How is the JS divergence in the invariance loss estimated? It is better to explicitly refer to the relevant section instead of saying "... is described below", which is confusing because it is not described immediately below. **
>
> JS divergence is estimated using a CNN-based architecture, whose details are provided in Appendix C.5 (JS-Divergence Estimators). We will update the cross-reference to avoid confusion.
>
> **L349: There are $N_1 \times ... \times N_P$ different $\mathbf n$'s, scaling quickly with the resolution and the group dimension. Is the summation over all possible $\mathbf n$'s efficient?**
>
> Our method relies on diagonalization based lifting, which substantially reduces the computational complexity. In this setting, the overall cost becomes approximately quadratic with respect to number of voxels for high batch sizes (when per-batch operations like diagonalization are neglected), which is comparable to standard matrix operations. We believe that this can be improved with matrix-factorization based approaches, which we plan to do in our future studies.
>
> Additional comment:
> **Then, from my understanding, the paper aims to discover a hidden space where the symmetry of the data distribution acts simply with a regular representation, i.e., shift invariance (wrt an Abelian group). The paper presents a convoluted way to achieve this, while I believe there are simpler solutions. For example, [1,2,3], among other methods for learning equivariant representations, can all be applied once we specify the group action on the hidden space (i.e., the symmetry-based domain in the context of this paper). Also, compared to the setting in this paper, these existing methods are not limited to (1) Abelian groups, and (2) data domains exactly being the group $\Omega=G$, but instead more general spaces equipped with a $G$-action. Therefore, a critical question is why the proposed method in this paper is superior to existing works in terms of applicability, efficiency, etc.**
>
> The key distinction of our method is its ability to operate directly on raw data and to recover real-world signals, not only to learn equivariant representations.
>
> Although methods such as [2,3] can discover equivariant representations, this alone is not sufficient for revealing the underlying real signals. In general, many different equivariant representations can describe the same entity, and the learned latent representation is unlikely to match the true generative signal. By imposing locality, we restrict the hypothesis space to a subset of equivariant representations, which allows the model to recover the actual real-world signals. Furthermore, our method does not require a sequential or structured representation to operate.
>
> Regarding [1], while the method is unsupervised, it relies on structured data (such as images), whereas our approach does not have this requirement.
>
> Compared to [3], our architecture does not require sequential or structured input representations. Our aim is to discover the most fundamental symmetries from raw observations and use them as a tool to assign structure to raw data / reveal the underlying signals.

---

### Official Review · Reviewer_wKZ4 · 2025-10-21

**Soundness:** 1
**Presentation:** 3
**Contribution:** 2
**Rating:** 4
**Confidence:** 3

**Summary:**

The paper proposes an unsupervised framework to discover symmetry groups and the associated hidden domains directly from raw data. The core idea is a lifting operation that maps inputs to a representation where the unknown symmetry acts as shifts. A lifting network learns commuting generators (restricted to abelian groups) and a resolving filter, while an auxiliary network estimates information theoretic quantities that drive the loss. Experiments on synthetic data show that the method can recover high‑dimensional group actions without paired or sequential samples.

**Strengths:**

S1. A novel setting for extracting symmetries without relying on sequence or paired data; independence from specific input structures is compelling.

S2. Compared to prior work, the method can learn higher-dimensional representations of group actions.

S3. The technical presentation is simple and easy to follow.

**Weaknesses:**

W1. Questionable realism of assumptions, especially the linear and invertible observation map M; many real tasks involve nonlinear or lossy transforms that may not permit full recovery.

W2. No experiments on real-world datasets; evaluation is limited to synthetic tasks.

W3. Practical utility is under-validated in the experiments:
- No comparisons against reasonable baselines.
- Loss ablations (in Appendix) are mostly qualitative; please provide quantitative ablations to show how much each term helps.

**Questions:**

Q1. Beyond permutation, what concrete real tasks fit your setting? Permuted MNIST does not feel realistic; examples from imaging, time series, or scientific data would help.

Q2. Why must the group be abelian? Where would the approach fail without commutativity, and what parts of the pipeline depend on it?

Q3. The bilevel/min–max training between the lifting and auxiliary networks could introduce instability (similar to GANs). Did you observe instability in practice? If so, what stabilization measures worked?

Q4. How were the loss weights alpha and beta selected?

Q5. For the resolution loss, what motivated its inclusion, and under what properties of the true signal does it help most? Since it promotes coordinate independence, when might it bias the reconstruction?

Minor comments
- The figure on page 6 appears without a figure number and caption; please add both. The text inside the panels is also hard to read at the current size.

---

> ### Author Response · Authors · 2025-12-03
>
> Dear reviewer,
>
> Thank you for your valuable feedback. We have decided to reorganize our manuscript for another conference.
>
> To address your questions:
>
> **Q1. Beyond permutation, what concrete real tasks fit your setting? Permuted MNIST does not feel realistic; examples from imaging, time series, or scientific data would help.**
>
> We are currently exploring real world applications. Discovering wave propagation operators from dispersive data could be a fruitful direction, as well as solving challenging inverse problems such as blind deconvolution.
>
> **Q2. Why must the group be abelian? Where would the approach fail without commutativity, and what parts of the pipeline depend on it?**
>
> As discussed In Section 3.2 Data Model and Signal Recovery, our hidden real world signals recovery result relies on applying group convolution (lifting convolution) under commutativity assumption. The derivation uses the fact that convolution is independent of element ordering, which allows us to identify a consistent and unique lifted representation.
>
> In the non-commutative case, this uniqueness becomes unclear: the lifted representation would depend on the ordering of group elements, making it difficult to assign a stable coordinate system to the underlying data. Additionally, the lifting procedure interacts with the group measure, and without commutativity, constructing a consistent representation becomes substantially more challenging.
>
> **Q3. The bilevel/min–max training between the lifting and auxiliary networks could introduce instability (similar to GANs). Did you observe instability in practice? If so, what stabilization measures worked?**
>
> We did not observe instability in practice, provided that the auxiliary network’s learning rate is kept an order of magnitude higher than that of the primary network. The auxiliary network consists of estimators for information-theoretic quantities, and our method requires these estimators to converge faster so that the primary network receives reliable gradients. Ensuring this faster convergence is considerably easier than maintaining a balance between two competing objectives, as in GANs, which likely contributes to the overall stability.
>
> **Q4. How were the loss weights alpha and beta selected?**
>
> All loss terms are formulated in terms of entropy and thus have comparable magnitudes by design; which simplifies hyperparameter tuning. We initially set all weights equal ($\alpha=\beta=1$), and this already yielded reasonable performance. In some configurations, however, we observed that assigning a slightly higher weight to the uniformity term improved stability, which motivated us to use configurations with $\alpha = \beta <1.0$. As shown in the sensitivity analysis in Appendix D.2, the influence of these hyperparameters is minimal overall, and there is large degree of freedom in choosing their values.
>
> **Q5. For the resolution loss, what motivated its inclusion, and under what properties of the true signal does it help most? Since it promotes coordinate independence, when might it bias the reconstruction?**
>
> In real-world datasets, symmetry operations often act on many small constituents independently, and the observed signal is a mixture of these components. To build a robust method, we believed that distinguishing these smaller components was important. Inspired by independent component analysis, we introduced total correlation minimization as resolution loss.
>
> As you mentioned, this does introduce a bias by coupling symmetry and locality. However, it also helps address the curse of dimensionality and makes it possible to recover hidden signals in higher dimensions—something that cannot be achieved by probabilistic-invariance alone.
>
> **The figure on page 6 appears without a figure number and caption; please add both. The text inside the panels is also hard to read at the current size.**
>
> Thank you for pointing this out, and we apologize for the inconvenience. We will improve our presentation.

---

### Official Review · Reviewer_16jy · 2025-10-30

**Soundness:** 2
**Presentation:** 2
**Contribution:** 2
**Rating:** 2
**Confidence:** 3

**Summary:**

This paper aims to discover symmetries of high dimensional data in an unsupervised way. Although there are other methods for discovering symmetry in data, this method is said to contribute in four key ways: (1) handling group actions which are not transitive; (2) discovery of symmetry-based domains; (3) scalability to higher dimensions; and (4) more generalized symmetry representations. Two networks are trained, namely a lifting network (leveraging group convolutions) and the so-called auxilliary network. The groups considered herein are matrix groups which are Abelian, and experiments deal with translations and frequency-shifting, among other types of invariance.

**Strengths:**

1. The notion of addressing scalability issues in symmetry detection is important, since many existing methods have been criticized for scalability problems.

2. This paper builds upon many interesting ideas, including group convolutions.

**Weaknesses:**

1. *Issue of transitivity discussion.* The authors are mistaken either in their understanding of transitive group actions or else in the group actions considered in recent literature. The authors claim that a novelty of their work is that their method works with intransitive group actions. However, a simple example of an intransitive group action is a group action in the plane defined by a rotation about the origin, which has most certainly been taken up by recent literature, much of which is cited by the authors. On the other hand, it is possible that the authors are misusing the term "transitive group action," as evidenced by the following statement (lines 407-413): "This setting forms a preliminary test to method’s ability for discovering symmetries when the dataset doesn’t involve identical samples those are globally translated versions of each other. In other saying, we aim to see if the model can learn the symmetries when the group acts intransitively." It seems that the authors may intend for the first sentence to serve as an implied definition of a transitive group action. However, if this is the case, the first statement does not provide an accurate definition of a transitive group action.

2. *The commutative case is quite restrictive.* In the plane, the set of rotations and translations are not commutative. Additionally, 3-d rotations are not generally commutative. The restriction to the commutative case is quite inhibitive, significantly affecting the extent to which this paper can be considered a meaningful contribution.

3. *The inabilities of other methods are not well established.* The authors claim that their method uniquely handles high-dimensional cases. But this should be demonstrated experimentally: in a high-dimensional case, the authors should show that all other methods (or at least a representative sample of other methods) either do not scale or fail to recover the ground truth symmetries. Experimental comparison could better illustrate the advantages of the proposed approach. Additionally, some existing methods do not require, as line 52 suggests, 100x100 matrices to capture translational symmetry in 10x10 data: see weakness 5 below.

4. *It is not clear what problems this method solves*. Weakness 3 discusses the issue with the claim of scalability to higher dimensions, and weakness 1 speaks to the claim of novelty regarding transitivity. But I also feel that contributions 2 and 4 are not sufficiently motivated: it is not clear what problems are benefited from these contributions. I acknowledge, however, that this perceived limitation could be a result of my own misunderstanding, and I welcome any clarity the authors can provide.

5. *Some work is left out of consideration*. The authors have not considered recent work on symmetry discovery using the Lie derivative, which may be of particular interest due to the focus on translational symmetry. Two recent papers are: (1) "Symmetry Discovery Beyond Affine Transformations" (Shaw et al., 2024), and (2) "A Unified Framework to Enforce, Discover, and Promote Symmetry in Machine Learning" (Otto et al., 2023).

6. *Experimental descriptions leave much to be desired*. There are several issues with the experimental descriptions. In particular, I do not know under which experiment to interpret Table 1. I think the operators of translation and frequency shifts need to be given explicitly, along with all other types of invariance which appear in Table 1. The meaning of the figures is not clear. Also, it's not entirely clear what the authors mean by a "translation-invariant dataset," given that the problem statement of finding a group of transformations which preserve the data distribution: truly, there are no translation invariant probability distributions. I also think the authors should relate each experiment to their stated contributions (which they do in the first experiment for intransitivity, which is problematic as mentioned in weakness 1). In fine, the experiments section seems as though it were placed in the paper as an afterthought: it lacks clarity, organization, and coherence.

**Questions:**

1. How does your computational complexity compare to other methods?

2. I'm not sure I understand how padding enforces translational invariance. Can you explain this?

3. Other questions are implied from the perceived weaknesses--in particular, weakness 6. Addressing the weaknesses may result in an increased score.

---

> ### Author Response · Authors · 2025-12-03
>
> Dear reviewer,
>
> Thank you for your valuable feedback, and pointing that we overlooked significant contributions. We have decided to reorganize our current manuscript for a different conference.
>
> To address your questions:
>
> **How does your computational complexity compare to other methods?**
>
> Our algorithm has computational complexity $O(d^3/B)$, where $B$ is batch size and $d$ is input dimensionality, which is similar to other raw-data–based methods such as SymmetryGAN and LieGAN. However, the default batch sizes used in GAN-based methods are significantly smaller than ours, which makes our method more efficient in practice.
>
> **I'm not sure I understand how padding enforces translational invariance. Can you explain this?**
>
> To ensure translation invariance, we need to sample patches whose spatial positions are equally likely. Without padding, patches near the image boundaries contain truncated content, while patches near the center contain fully observed content. This asymmetry introduces a position-dependent bias and breaks translation invariance.
>
> By padding the image before cropping, every possible crop location (including those that would fall partially outside the original image) produces a full and valid patch drawn from the same distribution. This ensures all spatial positions are equally represented and removes the boundary artifact that would otherwise violate translation invariance.
>
> **Other questions are implied from the perceived weaknesses—in particular, weakness 6. Addressing these weaknesses may result in an improved score.**
>
> We apologize that the experimental section was not sufficiently clear. Based on your feedback, we will revise the presentation accordingly.

---

### Official Review · Reviewer_vUGc · 2025-10-30

**Soundness:** 3
**Presentation:** 2
**Contribution:** 3
**Rating:** 6
**Confidence:** 4

**Summary:**

This paper presents a novel method for the unsupervised discovery of symmetries and symmetry-based domains from raw data. The key innovation is handling cases where the symmetry group acts intransitively on the dataset - meaning not all samples can be obtained from each other through symmetry operations. The approach uses a lifting operation inspired by Group Convolutional Networks (GCNs) to map observed signals to a representation space where symmetries manifest as simple translations. The method is demonstrated on various tasks including discovering pixel translations in MNIST images, recovering neighborhood structures in shuffled Ising models, and finding frequency-shift operators in time series data, achieving dimensionalities up to 27² × 27² for symmetry representations.

**Strengths:**

$\textbf{Originality}$: The handling of intransitive group actions is genuinely novel in unsupervised symmetry discovery. The connection to inverse problems through symmetry learning is creative.

$\textbf{Technical soundness}$: The theoretical framework connecting lifting operations, group representations, and information theory is well-developed. The use of the locality prior to enable scalability is clever.

$\textbf{Experimental diversity}$: Testing on translation, frequency-shift, permutation, and combined symmetries demonstrates generality
Practical impact: Real applications like unshuffling pixels (Figure 1) and recovering Ising model neighborhoods show the method's potential beyond toy problems.

The authors acknowledge limitations (abelian groups only, Gaussian distribution issues, performance degradation at scale) which is good.

**Weaknesses:**

$\textbf{Limited scalability analysis}$: While 27×27 is achieved, the drop in performance (Table 1, last row) suggests fundamental limitations. More analysis of how performance degrades with dimension would be valuable.

$\textbf{Comparison with baselines}$: The paper lacks quantitative comparisons with existing methods. Even if limited to 4×4, showing where your method excels would strengthen claims.

$\textbf{Gaussian distribution failure}$: This is mentioned but not thoroughly investigated. Since many real-world distributions are approximately Gaussian, this is concerning.

$\textbf{Hyperparameter sensitivity}$: The loss function has two hyperparameters (α, β) but no ablation study or guidance on setting them. How sensitive are results to these choices?

$\textbf{Computational cost}$: A detailed discussion of training time, memory requirements, or computational complexity compared to alternatives.

$\textbf{Non-abelian groups}$: While acknowledged as a limitation, some discussion of the fundamental barriers and potential paths forward would be valuable.

**Questions:**

Performance degradation: In Table 1, the 27×27 MNIST shows significantly lower cosine similarities (0.675-0.722). Is this a fundamental limitation or could it be addressed with more training/different architectures?

Hyperparameter selection: How were α and β chosen? What is the sensitivity to these values? Can you provide guidance for practitioners?

Gaussian distributions: You mention covariance-based joint entropy approximation as problematic for Gaussians. Have you tried alternative entropy estimators (e.g., k-NN based, kernel-based)?

Non-orthogonal representations: Section 5.1 mentions learning non-orthogonal distortions with a learnable K. How general is this? Could you elaborate?

Computational cost: What are the training times and memory requirements for different problem sizes?

Real inverse problems: Have you tested on any real inverse problems (e.g., image deblurring, compressed sensing)? This would greatly strengthen the contribution.

---

> ### Author Response · Authors · 2025-12-03
>
> Dear reviewer,
>
> Thank you for your kind words and valuable feedback. We have decided to reorganise our current manuscript for a different conference.
>
> To address your questions:
>
> **Performance degradation: In Table 1, the 27×27 MNIST shows significantly lower cosine similarities (0.675-0.722). Is this a fundamental limitation or could it be addressed with more training/different architectures?**
>
> We have discovered that TensorFlow eigensolver's numerical accuracy degrades significantly with the dimensionality of covariance matrix, which leads to poor joint entropy estimation and performance degradation. We plan to change it by more efficient and precise algorithm.
>
> **Hyperparameter selection: How were α and β chosen? What is the sensitivity to these values? Can you provide guidance for practitioners?**
>
> All loss terms are formulated in terms of entropy and thus have comparable magnitudes by design, which simplifies hyperparameter tuning. We initially set all weights equal ($\alpha=\beta=1$), and this already yielded reasonable performance. In some configurations, however, we observed that assigning a slightly higher weight to the uniformity term improved stability, which motivated us to use configurations with $\alpha = \beta <1.0$ As shown in the sensitivity analysis in Appendix D.2, the influence of these hyperparameters is minimal overall, and there is large degree of freedom in choosing their values.
>
> **Gaussian distributions: You mention covariance-based joint entropy approximation as problematic for Gaussians. Have you tried alternative entropy estimators (e.g., k-NN based, kernel-based)?**
>
> We did not explore these alternatives, as they typically have higher computational complexity compared to the Gaussian estimator. We favored the Gaussian approximation because it provides an efficient and robust way to estimate low-rank entropy. However, we plan to explore other estimators if we can implement them in a computationally efficient manner.
>
> **Non-orthogonal representations: Section 5.1 mentions learning non-orthogonal distortions with a learnable K. How general is this? Could you elaborate?**
>
> We are currently exploring possible applications. Our experiments imply that for a dataset with translation symmetry and spatial locality (for example, an image dataset), the model can restore the distorted representation. However, if the aim is to restore the original data and a symmetry generator, the distortion operation must be invertible therefore we limited our experiments to invertible distortions, whose singular values are limited in the range [0.5,2.0].
>
> **Computational cost: What are the training times and memory requirements for different problem sizes?**
>
> In Table 2 at Appendix B, we share the training times. In Appendix E, we share computational complexity. Current computational complexity is similar to matrix multiplication, which is $O(d^3/B)$ due to diagonalization algorithms which are executed for each batch. For high batch sizes, computational cost of shift-invariance estimation dominates, which is linear due to the CNN-based shift-invariance architecture.
>
> **Real inverse problems: Have you tested on any real inverse problems (e.g., image deblurring, compressed sensing)? This would greatly strengthen the contribution.**
>
> Thank you for the valuable suggestion. Although all our experiments can be regarded as synthetic inverse problems in a sense, we have not explored the real-world setting yet. We plan to include real world experiments in future work.

---

### Meta-Review · Area_Chair_J3nS · 2026-01-06

**Summary:**

1. needs more analysis on scalability (*vUGc*)
2. needs baseline comparisons (*vUGC*, *16jy*,*wKZ4*, *SCTN*)
3. why does the model fail on Gaussian distributions (*vUGc*)
4. should be sensitivity analysis for hyperparameters (*vUGc*, *16ky*)
5. needs a discussion of computational cost (*vUGc*)
6. the method is limited to abelian groups (which the authors acknowledge).   (*vUGc*, *16jy*) Overly strong assumptions of invertiblility and linearity. (*wKZ4*)
7. other authors have handled the case of intransitive group actions; the authors do not use the correct definition.(*16jy*, *SCTN*)
8. this paper does not address an actual weakness in existing methods; contributions are unclear (*16jy*, *SCTN*)
9. description of experiments is unclear (*16jy*, *SCTN*) no real world experiments (*wKZ4*)

**Reviewer Concerns:**

Note to the authors: Please feel free to respond to all parts of the reviews, not just the questions, and try to rebut or address the weaknesses.

1. unaddressed
2. unaddressed
3. The authors do not explore alternatives.
4. Analysis is in appendix.  There is a principled way to set their values.
5. The authors point to a discussion in the appendix
6. The authors already address this limitations explicitly.
7. unaddressed
8. The authors added explanation on how their method differs from previous methods.
9. The authors propose to clarify the experiment section.  They give examples of real world settings but do not perform experiments.

**Reviewer Scores:**

- *vUGc* would have kept their score of 6
- *16jy* would have kept their score of 2
- *wKZ4* would have kept their score of 4
- *SCTN* would have kept their score of 2

---

### Decision · Program_Chairs · 2026-01-26

Reject